# Data Quality in Imitation Learning

**Suneel Belkhale**
Stanford University
belkhale@stanford.edu

**Yuchen Cui**
Stanford University
yuchenc@stanford.edu

**Dorsa Sadigh**
Stanford University
dorsa@stanford.edu

## Abstract

In supervised learning, the question of data quality and curation has been over-shadowed in recent years by increasingly more powerful and expressive models that can ingest internet-scale data. However, in offline learning for robotics, we simply lack internet scale data, and so high quality datasets are a necessity. This is especially true in imitation learning (IL), a sample efficient paradigm for robot learning using expert demonstrations. Policies learned through IL suffer from state distribution shift at test time due to compounding errors in action prediction, which leads to unseen states that the policy cannot recover from. Instead of designing new algorithms to address distribution shift, an alternative perspective is to develop new ways of assessing and curating datasets. There is growing evidence that the same IL algorithms can have substantially different performance across different datasets. This calls for a formalism for defining metrics of "data quality" that can further be leveraged for data curation. In this work, we take the first step toward formalizing data quality for imitation learning through the lens of distribution shift: a high quality dataset encourages the policy to stay in distribution at test time. We propose two fundamental properties that shape the quality of a dataset: i) *action divergence:* the mismatch between the expert and learned policy at certain states; and ii) *transition diversity:* the noise present in the system for a given state and action. We investigate the combined effect of these two key properties in imitation learning theoretically, and we empirically analyze models trained on a variety of different data sources. We show that state diversity is not always beneficial, and we demonstrate how action divergence and transition diversity interact in practice.

## 1 Introduction

Supervised learning methods have seen large strides in recent years in computer vision (CV), natural language processing (NLP), and human-level game playing [18, 26, 43, 8, 51, 13, 44]. These domains have benefited from large and complex models that are trained on massive internet-scale datasets. Despite their undeniable power, biases present in these large datasets can result in the models exhibiting unexpected or undesirable outcomes. For example, foundation models such as GPT-3 trained on uncurated datasets have resulted in instances of racist behavior such as associating Muslims with violence [1, 5]. Thus offline data curation is immensely important for both safety and cost-effectiveness, and it is gaining in prominence in training foundation models [22, 10, 40].

Data curation is even more important in the field of robotics, where internet-scale data is not readily available and real-world datasets are small and uncurated. Noise or biases present in the data can lead to dangerous situations in many robotics tasks, for example injuring a person or damaging equipment. In such scenarios, deciding which data to collect and how best to collect it are especially important [37, 34, 4]. Of course, the quality of data depends on the algorithm that uses the data. A common paradigm in robot learning from offline datasets is imitation learning (IL), a data-driven, sample efficient framework for learning robot policies by mimicking expert demonstrations. However, when learning from offline data using IL, estimating data quality becomes especially difficult, since the "test set" the robot is evaluated on is an entirely new data distribution due to compounding errors

37th Conference on Neural Information Processing Systems (NeurIPS 2023).

incurred by the model, i.e., action prediction errors take the model to unseen states. This phenomenon is studied and referred to as the distribution shift problem, and prior work has viewed and addressed it through several angles [46, 49].

Broadly, prior work address distribution shift either by taking an *algorithm-centric* approach to account for biases in the dataset, or by directly modifying the dataset collection process in a *data-centric* manner. Algorithm-centric approaches learn robust policies by imposing task-specific assumptions [23, 28, 25], acquiring additional environment data [21, 41], or leveraging inductive biases in representing states [54, 30, 38, 12, 12, 37] and actions [53, 48, 12]. What these algorithm-centric works overlook is that changing the data can be as or more effective for policy learning than changing the algorithm. In prior data-centric methods, the goal is usually to maximize *state diversity* through shared control [46, 45, 16, 27, 32, 36, 47], noise injection [34], or active queries [15, 6, 42, 29]. By only focusing on state coverage, these works are missing the role that actions (i.e., the expert) plays in the quality of data. A more complete understanding of data quality that integrates both state and action quality would not only improve performance but also save countless hours of data collection.

To better understand the role of both states and actions in the data for learning good policies, consider a single state transition — the three core factors that affect distribution shift are: the policy action distribution, the the stochasticity of transitions, and the previous state distribution. Note that the previous states are also just a function of the policy and dynamics back through time. Through this we extract two fundamental properties that can influence data quality in IL: *action divergence* and *transition diversity*. Action divergence captures how different the learned policy action distribution is from the expert's actions at a given state: for example, if the expert is very consistent but the policy has high variance, then action divergence will be high and distribution shift is likely. Transition diversity captures the inherent variability of the environment at each state, which determines how the state distribution evolves for a given policy: for example, noisy dynamics in the low data regime can reduce overlap between the expert data and the learned policy distribution. Importantly, these factors can compound over time, thus greatly increasing the potential for distribution shift. While state coverage has been discussed in prior work, we are the first work to formalize the roles of both state and action distributions in data quality, along with how they interact through time: this new data-focused framing leads to insights about how to curate data to learn more effective policies.

To validate and study these properties empirically, we conduct two sets of experiments: (1) Data Noising, where we ablate these properties in robotics datasets to change the policy success rates, and (2) Data Measuring, where we observe human and machine generated datasets and approximately measure these properties in relationship to the policy success rates. Both experiments show that state diversity, a commonly used heuristic for quality in prior work, is not always correlated with success. Furthermore, we find that in several human generated datasets, less consistent actions at each state is often associated with decreases in policy performance.

## 2 Related Work

Data quality research dates back to the early time of computing and the full literature is out of the scope of this paper. Existing literature in machine learning has proposed different dimensions of data quality including accuracy, completeness, consistency, timeliness, and accessibility [33, 14, 20]. We could draw similarities between action divergence with the concept of data consistency, and state diversity with completeness. However, instantiation of these metrics is non-trivial and domain-specific.

Our work is closely related to the imitation learning literature that explicitly addresses the distribution shift problem in various ways. Prior work can be divided into two camps: ones that take an *algorithm-centric* approach to account for biases in the dataset, and ones that employ *data-centric* methods for modifying the data collection process.

**Algorithm-centric.** Robust learning approaches including model-based imitation learning methods learn a dynamics model of the environment and therefore can plan to go back in distribution when the agent visits out of distribution states [21, 41]. Data augmentation is a useful post-processing technique when one could impose domain or task specific knowledge [23, 28, 25]. Prior work has also investigated how to explicitly learn from sub-optimal demonstrations by learning a weighting function over demonstrations either to guide BC training [9, 3] or as a reward function for RL algorithms [7, 11]. These methods either require additional information such as rankings of trajectories [9], demonstrator identity [3], or collecting additional environment data [7, 11]. Recent efforts on demonstration retrieval augment limited task-specific demonstrations with past robot experiences [39]. A recent

body of work build better inductive biases into the model for learning robust state features, like pretrained vision or language representations [54, 30, 38, 12]. Some approaches modify the action representation to be more expressive to capture all the expert's actions, for example using Gaussian or mixture model action spaces [12, 37]. Others consider temporally abstracted action spaces like waypoints or action sequences to reduce the effective task length and thus mitigate compounding errors [53, 48, 12]. What these algorithm-centric works overlook is that changing the data can be as or more effective for policy learning than changing the algorithm. However, we still lack a comprehensive understanding of what properties in the data matter in imitation learning.

**Data-centric.** In more data-centric prior works that discuss data quality, the primary goal is often just to maximize state diversity. A large body of research focuses on modifying the data collection process such that the expert experience a diverse set of states. Ross et al. [46] proposed to iteratively collect *on-policy* demonstration data with shared-control between the expert and the robot, randomly switching with a gradually decreasing weight on the expert's input, such that the training data contains direct samples of the expert policy at states experienced by the learned policy. However, randomly switching the control can make it unnatural for the human demonstrator and leads to noisier human control. To mitigate this issue, methods have been proposed to gate the control more effectively by evaluating metrics such as state uncertainty and novelty [16, 27, 50]. Other methods allow the human to gate the control and effectively correct the robot's behavior only when necessary [32, 36, 24, 47]. Closely related to our work, Laskey et al. [34] takes an optimal control perspective and showed that injecting control noise during data collection can give similar benefits as DAgger-style iterative methods. Active learning methods have also been developed to guide data collection towards more informative samples [15, 6, 42, 29]. By only focusing on state coverage, these works are missing the role that actions (i.e., the expert) plays in the quality of data.

## 3 Preliminaries

In imitation learning (IL), we assume access to a dataset $\mathcal{D}_N = \{\tau_1, \ldots, \tau_N\}$ of $N$ successful demonstrations. Each demonstration $\tau_i$ consists of a sequence of continuous state-action pairs of length $T_i$, $\tau_i = \{(s_1, a_1), \ldots, (s_{T_i}, a_{T_i})\}$, with states $s \in \mathcal{S}$ and actions $a \in \mathcal{A}$. Demonstrations are generated by sampling actions from the demonstration policy $\pi_E$ through environment dynamics $\rho(s'|s, a)$. The objective of imitation learning is to learn a policy $\pi_\theta : \mathcal{S} \to \mathcal{A}$ that maps states to actions. Standard behavioral cloning optimizes a supervised loss maximizing the likelihood of the state-action pairs in the dataset:

$$\mathcal{L}(\theta) = -\frac{1}{|\mathcal{D}_N|} \sum_{(s,a) \in \mathcal{D}_N} \log \pi_\theta(a|s), \tag{1}$$

which is optimizing the following objective under finite samples from the demonstration policy:

$$\mathbb{E}_{s \sim \rho_{\pi_E}(\cdot)} \left[ D_{\text{KL}} \left( \pi_E(\cdot|s), \pi_\theta(\cdot|s) \right) \right] = -\mathbb{E}_{s \sim \rho_{\pi_E}(\cdot), \, a \sim \pi_E(\cdot|s)} \left[ \log \pi_\theta(a|s) \right] + C \tag{2}$$

The $C$ term here captures the entropy of the demonstration state-action distribution, which is constant with respect to $\theta$ and thus it does not affect optimality of $\theta$. $\rho_{\pi_E}(s)$ is the state visitation of the demonstration policy, defined for any policy $\pi$ as follows:

$$\rho_\pi(s) = \frac{1}{T} \sum_{t=1}^{T} \rho_\pi^t(s) \tag{3}$$

$$\rho_\pi^t(s') = \int_{s,a} \rho_\pi^t(s, a, s') ds \, da = \int_{s,a} \pi(a|s) \rho(s'|s, a) \rho_\pi^{t-1}(s) ds \, da \tag{4}$$

### 3.1 Distribution Shift in IL

Behavioral cloning methods as in Eq. (2) often assume that the dataset distribution is a good approximation of the demonstration policy distribution. In most applications, however, the learned policy is bound to experience novel states that were not part of the training data due to stochasticity of the environment dynamics and the learned policy. Herein lies the fundamental challenge with imitation learning, i.e., state *distribution* shift between training and test time. Consider the training sample $(s, a, s')$ at timestep $t$ in demonstration $\tau_i$. If the learned policy outputs $\tilde{a} \sim \pi(\cdot|s)$ which has a small action error $\epsilon = \tilde{a} - a$, the new state at the next time step will also deviate: $\tilde{s}' \sim \rho(s'|s, a + \epsilon)$, which in turn affects the policy output at the next step. In practice, the change in next state can be

highly disproportionate to $||\epsilon||$, so small errors can quickly lead the policy out of the data distribution. Stochastic dynamics, for example system noise, can compound with the distribution shift caused by policy errors, and as we continue to execute for $T - t$ steps, both these factors can compound over time to pull the policy out of distribution, often leading to task failure.

We can address this distribution shift problem by minimizing the mismatch between state visitation distribution of a learned policy $\pi$ and the state visitation distribution of an demonstration policy $\pi_E$ under some $f$-divergence $D_f$:

$$J(\pi, \pi_E) = D_f(\rho_\pi(s), \rho_{\pi_E}(s)) \tag{5}$$

Next, we connect this distribution shift problem to the question of data quality.

## 3.2 Formalizing Data Quality

How can we define and measure the quality of a dataset $\mathcal{D}$ in IL? In existing literature, data quality has been used interchangeably with either the proficiency level of the demonstrator at the task or the coverage of state space. However, these notions of quality are loose and incomplete: for example, they do not explain what concretely makes a demonstrator better or worse, nor do they consider how stochasicity in the transitions impacts state space coverage and thus downstream learning. Our goal is to more formally define a measure of quality so that we can then optimize these properties in our datasets.

We posit that a complete notion of dataset quality is one that minimizes distribution shift in Eq. (5). This suggests that we cannot discuss data quality in isolation. The notion of data quality is heavily tied to the algorithm $A$ that leads to the learned policy $\pi_A$ as well as the demonstration policy $\pi_E$. We thus formalize the quality of a dataset $\mathcal{D}_N$ conditioned on the demonstration policy $\pi_E$ that generates it along with a policy learning algorithm $A$, as the negative distribution shift of a learned policy under some $f$-divergence $D_f$:

$$Q(\mathcal{D}_N; \pi_E, A) = -D_f\left(\rho_{\pi_A}(s), \rho_{\pi_E}(s)\right), \text{ where } \pi_A = A(\mathcal{D}_N) \tag{6}$$

Here, $\pi_A$ is the policy learned by algorithm $A$ using dataset $\mathcal{D}_N$ of size $N$, which is generated from demonstrator $\pi_E$. Note that quality is affected by several components: the choice in demonstrator, the choice of dataset size, and the algorithm. The choice of demonstration policy $\pi_E$ changes the trajectories in $\mathcal{D}_N$ (which in turn affects policy learning) along with the desired state distribution $\rho_{\pi_E}(s)$. The dataset size controls the amount of information about the demonstration state distribution present in the dataset: note that $\rho\pi_A(s)$ should match $\rho_{\pi_E}(s)$, but $A$ learns only from $\mathcal{D}_N$ not from the full $\rho_{\pi_E}(s)$. The algorithm controls how the data is processed to produce the final policy $\pi_A$ and thus the visited distribution $\rho_{\pi_A}(s)$.

To optimize this notion of quality, we might toggle any one of these factors. Prior work has studied algorithm modifications at length [37, 54, 4], but few works study how $\pi_E$ and the dataset $D_N$ should be altered to perform best for any given algorithm $A$. We refer to this as *data curation*. In practice we often lack full control over the choice of $\pi_E$, since this is usually a human demonstrator; still, we can often influence $\pi_E$, for example through prompting or feedback [24], or we can curate the dataset derived from $\pi_E$ through filtering [19].

## 4 Properties of Data Quality

To identify properties that affect data quality in imitation learning, we can study how each state transition $(s, a, s') \in \mathcal{D}$ in the dataset affects the dataset quality in detail. As we defined in Eq. (6), the dataset quality relies on distribution shift given an demonstration policy $\pi_E$ and an imitation learning policy $\pi_A$. This relies on the state visitation distribution $\rho_{\pi_A}^t(s)$, which based on Eq. (4) depends on three terms: $\pi_A(a|s)$, $\rho(s'|s, a)$, and $\rho_{\pi_A}^{t-1}(s)$. Intuitively, three clear factors emerge for managing distribution shift: how different the policy $\pi_A(a|s)$ is from the demonstrator $\pi_E(a|s)$ – which we refer to as *action divergence*, how diverse the dynamics $\rho(s'|a, s)$ are – which we refer to as *transition diversity*, and how these factors interact over time to produce past state visitation distribution $\rho_{\pi_A}^{t-1}(s)$. Importantly, the past state visitation can be controlled through *action divergence* and *transition diversity* at previous time steps, so in this section, we will formalize these two properties and discuss their implication on data curation.

## 4.1 Action Divergence

Action divergence is a measure of distance between the learned policy and the demonstration policy $D_f(\pi_A(\cdot|s), \pi_E(\cdot|s))$. This can stem from biases in the algorithm or dataset such as mismatched action representations or lack of samples. While this is typically viewed in prior work as a facet of the algorithm or dataset size, importantly action divergence and thus data quality can also be influenced by the demonstration policy itself. For example, if the demonstrator knows the action representation used by the learning agent *a priori*, then the policy mismatch can be reduced by taking actions that are consistent with that action space. In Theorem 4.1[1], we illustrate the importance of action divergence by showing that our notion of data quality is lower bounded by the action divergence under the visited state distribution.

**Theorem 4.1.** *Given a policy $\pi_A$ and demonstrator $\pi_E$ and environment horizon length $H$, the distribution shift:*

$$D_{KL}(\rho_{\pi_A}, \rho_{\pi_E}) \leq \frac{1}{H} \sum_{t=0}^{H-1} (H-t) \mathbb{E}_{s \sim \rho_{\pi_A}^t} \left[ D_{KL}(\pi_A(\cdot|s), \pi_E(\cdot|s)) \right]$$

When using KL divergence for distribution shift in Eq. (5), we can see how the quality is lower bounded by the policy action divergence from the optimal policy under the visited state distribution, weighted at each step by time-to-go. Many prior works have noted the compounding error problem [46, 45], and here too action divergence at earlier timesteps has an out-sized effect on the overall distribution shift. Thus to optimize for the quality of a dataset, we should reduce the aggregated policy mismatch across visited trajectories.

**Optimality**: Many prior ideas around the "optimality" of a demonstration are intimately related to this action divergence property. Suboptimalities that are naturally present in demonstrations like pauses or other action noise have been a source of difficulty when learning imitation policies [35, 4]. These factors make it harder for the model to learn the demonstration action distributions, thus increasing action divergence. Suboptimality can also come from multi-modal demonstration policy distributions, and more expressive policy representations (i.e., reducing action divergence between the demonstrator and the learned policy) has been shown to help [17, 12, 24]. The speed of demonstration is another common notion of optimality, which decreases total action divergence by reducing horizon length $H$, provided it does not increase the per time step action divergence to compensate.

**State Visitation**: Critically, the visitation distribution in Theorem 4.1 determines *where* action divergence should be low, whereas in the standard BC objective in Eq. (2), the divergence is only minimized under samples from the demonstration distribution $\rho_{\pi_E}^t$. To better understand how the visitation distribution evolves, we now analyze state transitions in greater detail.

## 4.2 Transition Diversity

Transition diversity encompasses the diversity of next state transitions seen at a given state for a certain policy. What role does transition diversity play in minimizing distribution shift? Intuitively, if we consider the upper bound in Theorem 4.1, the demonstrator's state coverage should overlap as much as possible with the visited state distribution, but without increasing the action divergence.

**Lemma 4.2.** *Given a learned policy $\pi_A$ and an demonstration policy $\pi_E$, assume that the policy is learned such that when $s \in \text{supp}(\rho_{\pi_E}^t)$, $D_{KL}(\pi_A(\cdot|s), \pi_E(\cdot|s)) \leq \beta$. Then:*

$$\mathbb{E}_{s \sim \rho_{\pi_A}^t} \left[ D_{KL}(\pi_A(\cdot|s), \pi_E(\cdot|s)) \right] \leq \mathbb{E}_{s \sim \rho_{\pi_A}^t} [\beta \mathbb{1}(s \in \text{supp}(\rho_{\pi_E}^t))$$
$$+ \mathbb{1}(s \notin \text{supp}(\rho_{\pi_E}^t)) D_{KL}(\pi_A(\cdot|s), \pi_E(\cdot|s))]$$

Lemma 4.2 shows that with a good learning algorithm (i.e., when $\beta$ is small), it is important to maximize the overlap between the visited policy and the demonstration data (minimizes the contribution of the unbounded right term), but as shown in Theorem 4.1 this should not come at the expense of increasing the action divergence of the policy. Rather than broadly maximizing state diversity, as prior works do, a more sensible approach is for the demonstrator to maximize the two other factors that affect $\rho_{\pi_A}^t$: *system noise* and *initial state variance* in the demonstrator data.

---

[1]Proof for all theorems and lemmas in Appendix A

**Finite Data and Low Coverage**: While the above analysis holds for infinite data, maximizing transition diversity can lead to thinly spread coverage in the finite data regime. To analyze the finite data regime in isolation, we consider Gaussian system noise for simplicity. First, we formulate an empirical estimate of state "coverage" of the demonstrator (i.e. the deviation from states in the demonstration dataset). We then show that higher system noise can cause the learned policy $\pi_A$ to deviate more from states sampled in the dataset.

**Definition 1.** *For a given start state $s$, define the next state coverage probability over $N$ samples from $\pi_E$ for tolerance $\epsilon$ as $P_S(s; N, \epsilon) \coloneqq P(\min_{i \in \{1...N\}} ||s' - s'_{*,i}||_\infty \leq \epsilon)$, where $\{s'_{*,i} \sim \rho^t_{\pi_E}(\cdot|s)\}$ is a set of $N$ sampled next state transitions under the demonstration policy $\pi_E$ starting at $s$, and $s' \sim \rho^t_{\pi_A}(\cdot|s)$ is a sampled next state under the policy $\pi_A$ starting at $S$.*

We can think of $P_S(s; N, \epsilon)$ as the probability under potential datasets for $\pi_E$ of seeing a next state at test time (under $\pi_A$) that was nearby the next states in the dataset, conditioned at a starting state $s$ (i.e., coverage). This is related to a single step of distribution shift, but measured over a dataset rather than a distribution. Defining coverage as the distance to the dataset being below some threshold is reasonable for function approximators like neural networks [2].

**Theorem 4.3.** *Given a policy $\pi_A$ and demonstrator $\pi_E$, assume that for state $s$, if $\rho^{t-1}_{\pi_E}(s) > 0$, then $\pi_A(a|s) = \pi_E(a|s)$. Assume that transitions are normally distributed with fixed and diagonal variance, $\rho(s'|s, a) = \mathcal{N}(\mu(s, a), \sigma^2 I)$, then the next state coverage probability is $P_S(s; N, \epsilon) \geq 1 - \left(1 - (\frac{c\epsilon}{\sigma})^d \exp\left(-\alpha^2 d\right)\right)^N - \exp\left(-(\alpha-1)^2 d\right)$, where $d$ is the dimensionality of the state, $c$ is a constant, and $\alpha \geq 1$ is a parameter chosen to maximize the bound.*

In Theorem 4.3, we see that even under a policy that is perfect when in distribution, the lower bound of next state coverage probability increases as the variance of system noise $\sigma^2$ decreases, for a fixed sampling budget $N$ (and fixed $\alpha$). However, increasing $N$ has a much stronger positive effect on this lower bound of state coverage than decreases in $\sigma$, suggesting that system noise is only an issue when the dataset size is sufficiently small. Furthermore, $\epsilon$ here represents some simplistic form of the *generalization* capacity of the policy, and we see here that increasing $\epsilon$ makes us more robust to proportional increases in $\sigma$.

**Finite Data and Good Coverage**: However, if we assume $N$ is large enough to mitigate the coverage effect of system noise, are there any other effects of transition diversity in the dataset? The answer lies in generalization properties of the algorithm, which as provided in Theorem 4.3 are fairly conservative (i.e., in the definition of $P_S(s; N, \epsilon)$). In Theorem A.1 in Appendix A, we use a more loose generalization definition and relax the assumption that the learned policy is perfect. We show as system noise increases, the resulting boost in sample coverage can actually replicate and overcome the effects of high learned policy noise, suggesting that more system noise can actually be beneficial for learning. This finding sheds light on results from prior work and our own experiments in Section 5 that show the benefits of collecting data in the presence of high system noise [34].

### 4.3 Implications for Data Curation

We have shown how action divergence and transition diversity are both tied to distribution shift and thus data quality. Based on action divergence and transition diversity, we now examine downstream implications of these properties on what matters for *data curation*, where the goal is to collect and then select *high quality* demonstrations in our dataset $\mathcal{D}_N$ for good policy learning.

**Action Consistency**: To minimize action divergence, the algorithm action representation should *align* with the demonstrator's action distribution, for the given dataset size. One algorithmic solution is to improve the expressiveness of the policy action space so it can capture the exact actions at every state that was demonstrated. However, in practice, individual states might only be visited a few times in the dataset, and so if the entropy of the demonstration policy at those states is high, the learned policy will find it difficult to perfectly match the actions even for the most expressive action spaces. Instead, we argue that demonstration data should be curated to have more *consistent* actions, e.g., reducing the entropy of the demonstration policy: $\mathbb{E}_{s \sim \rho_{\pi_A}(\cdot)}[\mathcal{H}(\pi_E(\cdot|s))]$. Since knowing the visited state distribution is impossible beforehand, the best we can do is to encourage low entropy in the demonstration data distribution:

$$\min_{\pi_E} \mathbb{E}_{s \sim \rho_{\pi_E}(\cdot)}[\mathcal{H}(\pi_E(\cdot|s))] \tag{7}$$

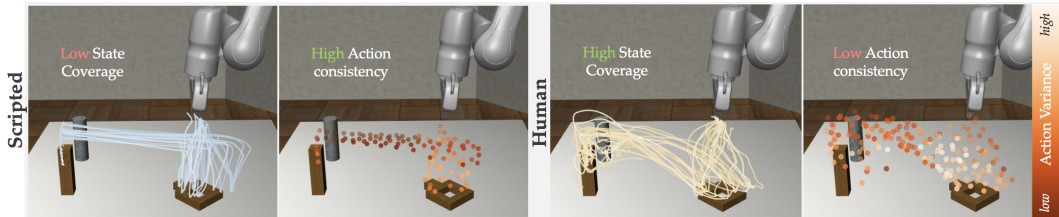

Figure 1: **Case Study**: Trajectories and action variance for scripted (left two plots) compared to human demonstration data (right two plots). Even though the human data (right) has high state coverage, the action variance is high, leading to high action divergence, and vice versa.

**State Diversity**: As discussed previously, the state visitation of the policy depends on the trajectory action distribution, the initial state distribution, and the system noise distribution. Many prior works seek to improve the *state coverage* of the dataset, using some metric similar to Definition 1 [6, 29]. However, the required state diversity is a function of the learned policy mismatch (action divergence) and the noise present in the system (transition diversity), and is thus a coarse representation of data quality. Uniformly increasing coverage over the state space is not necessarily a good thing — as shown in Section 4.1, if state coverage comes at the expense of action consistency, the result might be worse than a policy trained on less state coverage but more consistent actions. Instead, we argue that we should optimize for system noise.

**System Noise**: Although system noise at a given state cannot be controlled, demonstrators can control system noise at the trajectory level (e.g. visiting states with more or less system noise). Should system noise be encouraged or discouraged in a dataset? Based on the discussion in Section 4.2, we hypothesize that when the dataset size is fixed, increasing the system noise can increase the *coverage* of the demonstrations and thus improve robustness in the learned policy (Theorem A.1), but only up to a certain point, when the data becomes too sparse to generalize (Theorem 4.3). Thus, in addition to consistent actions, we posit that demonstrators should encourage paths with high system entropy for learning better policies, such that the overall state entropy stays below some threshold $\gamma$ to avoid coverage that is too sparse.

$$\max_{\pi_E} \mathbb{E}_{s \sim \rho_{\pi_E}(\cdot), a \sim \pi_E(\cdot|s)}[\mathcal{H}(\rho(\cdot|a, s))]$$
$$s.t. \ H(\rho_{\pi_E}(s)) \leq \gamma \tag{8}$$

**Horizon Length**: The length of trajectories in the demonstration data also can have a large effect on the demonstration and visited state distributions, and thus on the quality of the dataset. However, while horizon length certainly plays a role, like state diversity, it is a downstream effect of action divergence and transition diversity. Based on previous analysis, what really matters is minimizing the *aggregated* action divergence and transition diversity produced by the demonstrator across time. Horizon length alone only crudely measures this, but is often correlated in practice as we show in Section 5.2.

While prior work in data curation primarily aim to maximize state diversity alone, our analysis of data quality reveals several new properties that matter for data curation, such as action consistency, system noise, and their combined effect over time. As we demonstrate in the next section, understanding these properties and the inherent tradeoffs between them is vital for data curation in imitation learning.

# 5 Analysis

To empirically analyze how different data properties affect imitation learning, we conduct two sets of experiments. In the first set, we generate various quality demonstration datasets by adding different types of noise to *scripted* policies, i.e., policies that are hand-designed to be successful at the task. An example scripted policy compared to human demonstrations is shown in Fig. 1. We study both noise in the system – considering transition diversity – and noise added to the demonstration policy – considering both transition diversity and action divergence – as a function of dataset size. In the second set of experiments, we evaluate the empirical properties (see Appendix B) for real human collected datasets.

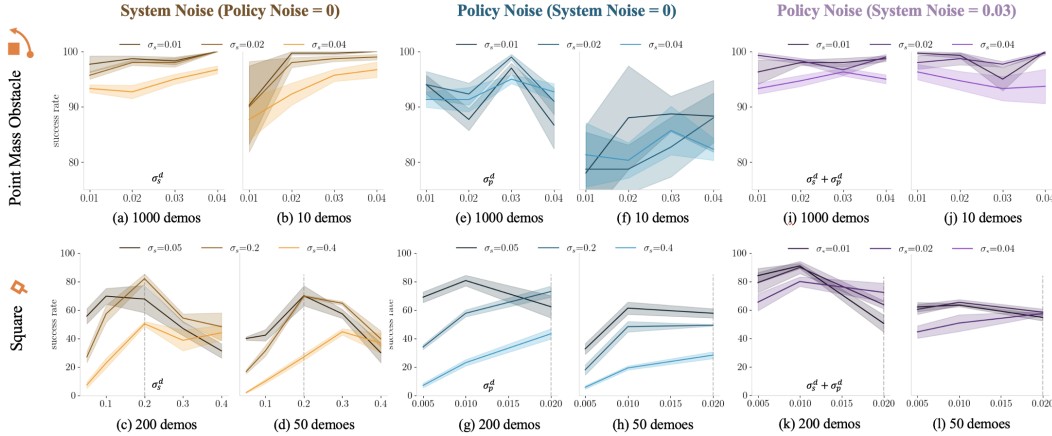

Figure 2: BC Success rates in *PMObstacle* (top row) for 1000 and 10 episodes of data, and in *Square* (bottom row) for 200 and 50 episodes of data (error bars over 3 datasets). X-axis corresponds to injected system or policy noise in the *dataset* ($\sigma_s^d$ or $\sigma_p^d$) and each line corresponds to injected noise ($\sigma_s$ or $\sigma_p$) during *evaluation*. **System Noise** (left): for large datasets (a)(c), higher system noise during evaluation decreases performance, but more system noise during training does the best. For small datasets (b)(d), we note a similar but exaggerated effect. **Policy Noise** (mid) For large datasets (e)(g), unlike system noise, more demonstration policy noise often hurts performance despite similar state coverage. For small datasets (f)(h), adding policy noise exaggerates this effect and produces large variance in performance. For *Square*, the dotted lines mark comparable values of noise in terms of state coverage. **System + Policy Noise** (i-l): Adding some system noise can make the policy more robust to action divergence.

## 5.1 Data Noising

We train Behavior Cloning (BC) with data generated with system noise and policy noise in two environments: *PMObstacle*, a 2D environment where a point mass agent must reach a target point without hitting an obstacle in its way, where demonstrations are multimodal (left and right around the obstacle); and *Square*, a 3D environment where a robot arm must pick up a square nut and insert it onto a peg (shown in Fig. 1). In both, system and policy noise are Gaussian random variables added to the dynamics and scripted policy, respectively, at each time step, and BC uses an MLP architecture. Fig. 2 shows results in *PMObstacle* (top row) and *Square* (bottom row). We additionally show Transformer architecture results in Fig. 3.

**State Diversity through Transition Diversity improves performance, up to a point**. The left plots in Fig. 2 (a)-(d) show policy success rates as we vary system noise ($\sigma_s$) (more noise for lighter color shades), where (a)(c) show the high-data regime and the (b)(d) show the low data regime. Higher system noise tends to improve policy performance in the high data regime, but only up to a point for *Square* — after $\sigma_s = 0.3$ in the demonstration data (c), performance starts to drop off, which we hypothesize is due to the low coverage, based on the analysis of transition diversity in Section 4.2. In the low data regime, the story is similar but even more exaggerated, with increasing system noise leading to comparable performance as the high data regime. Once again for *Square*, increasing transition diversity (d) helps until the state coverage is too thin. See Table 2, Table 5, and Table 6 in Appendix C for the full sweep of system noise values in each environment.

**State Diversity at the cost of Action Divergence hurts performance**. Plots in Fig. 2 (e)(f) and Fig. 2 (i)(j) show policy success rates for increasing the policy noise ($\sigma_p$) in the dataset, where the (e)(g) show the high data regime and the (f)(h) show the low data regime. Note that each value of policy noise yields the same demonstration state distribution as the corresponding amount of system noise. Since this noise is zero-mean and the learned policy is deterministic, policy noise in the high data regime (e)(g) is only moderately worse as compared to equivalent amounts of system noise (a)(c), suggesting that the action divergence is minor. In fact, due to the added state diversity, adding policy noise in the high data regime helps up to a certain point. However in the low data regime (f)(h), performance is substantially worse compared to comparable system noise, since the policy can not recover the unbiased demonstration policy from just a few noisy examples (note that in (c)(d), the x-axes are not aligned with those in (g)(h), and $\sigma_s = 0.2$ corresponds to $\sigma_p = 0.02$ as shown by the dotted line). This illustrates that state diversity coming from the noise in the demonstration policy

can increase action divergence, and thus high state diversity is not universally desirable. See Table 3, Table 7, and Table 8 in Appendix C for the full sweep of policy noise values in each environment.

**Transition Diversity can counteract the effects of policy noise**. In the right two plots (i)(j) for *PMObstacle* in Fig. 2, the dataset combines both system and policy noise. The system noise is fixed ($\sigma_s = 0.03$) and the policy noise is varied in the same range as the middle two plots (just policy noise). Given the analysis in Section 4.2 and Theorem A.1 in Appendix A, we would expect that having some system noise could actually make the policy much more robust to policy noise, since the state coverage provided in the dataset can help learn a more robust policy (generalization). Indeed we find that just by adding system noise, the learned policy becomes very robust to added policy noise in both the high (i) and low (j) regimes. This suggests that adding transition diversity can help mitigate the affects of action divergence incurred by noisy or sub-optimal demonstrators. See Table 4 in Appendix C for the full sweep of policy noise values for fixed system noise in each environment.

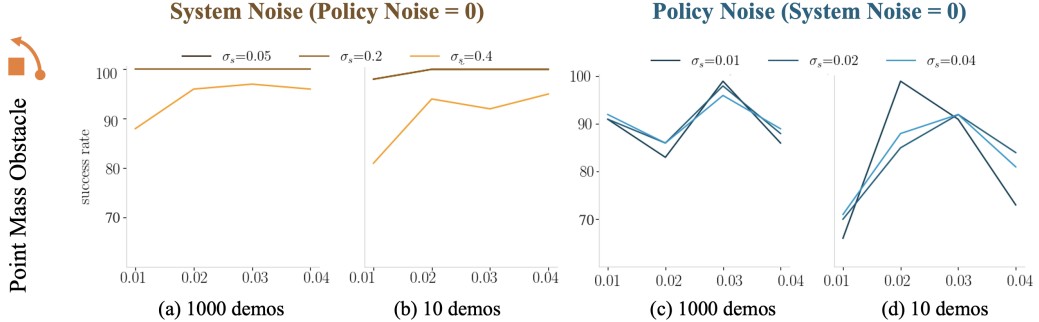

Figure 3: Transformer-BC Success rates in *PMObstacle* for 1000 and 10 episodes. X-axis corresponds to injected system or policy noise in the *dataset* and each line corresponds to injected noise ($\sigma_s$ or $\sigma_p$) during *evaluation*. **System Noise** (left): for large datasets (a), higher system noise during evaluation decreases performance, but more system noise during training does the best. For small datasets (b), we note a similar but exaggerated effect. **Policy Noise** (mid) For large datasets (c), unlike system noise, more demonstration policy noise often hurts performance despite Transformers being more expressive than simple MLPs. For small datasets (d), adding policy noise exaggerates this effect and produces large variance in performance. Overall, we find that Transformers can suffer from similar data quality issues as MLPs despite being more expressive.

## 5.2 Data Measuring

Next we empirically instantiate several metrics from Section 4.3 that capture various facets of data quality, and measure different datasets: (1) action variance (higher means less action consistency) measured through clustering nearby states (2) horizon length $T$ (higher means longer trajectories), and (3) state similarity (opposite of state diversity) measured as the average cluster size. See Appendix B for the exact empirical definitions. We leave out system noise since these human datasets were collected in deterministic environments.

In Table 1, we consider single and multi-human datasets from the *Square* and *Can* tasks from robomimic [37]. We see that higher action variance and horizon length are often accompanied by decreases in success rates. The *Worse* multi-human datasets have seemingly lower action variance but less state similarity, and yet qualitatively we observed highly multi-modal behaviors in these datasets (e.g., grasping different parts of the nut instead of the handle). We suspect that this type of multi-modality is not measured well by the single time step action variance metric, but rather requires a longer term view of the effects of that action on the scene – as indicated in Theorem 4.1. Additionally, state diversity once again is not correlated with performance on the task.

We emphasize that these metrics likely do not provide a complete picture of data quality, but they do provide us insights into why performance is low and how data collection can be improved in future datasets. Through future efforts from the community, we envision a comprehensive set of data metrics that practitioners can use to quickly evaluate the quality of datasets without needing to first train and evaluate models on those datasets.

|  | *Square* | | | | *Can* | | | |
|---|---|---|---|---|---|---|---|---|
|  | PH | Better | Okay | Worse | PH | Better | Okay | Worse |
| Success Rate | 58 | 36 | 12 | 2 | 96 | 56 | 40 | 22 |
| Dataset Size ($N$) | 200 | 100 | 100 | 100 | 200 | 100 | 100 | 100 |
| Action Variance | 0.073 | 0.062 | 0.099 | 0.061 | 0.051 | 0.066 | 0.079 | 0.063 |
| Horizon Length ($T$) | 150 | 190 | 250 | 350 | 115 | 140 | 180 | 300 |
| State Similarity | 8.2e-5 | 1.8e-4 | 1.7e-4 | 1.2e-4 | 1.0e-4 | 2.1e-4 | 2.4e-4 | 2.0e-4 |

Table 1: Data Quality metrics evaluated on *Square* (left) and *Can* (right) for proficient human (PH), and multi human (Better, Okay, Worse) datasets. Note that state similarity (defined in Appendix B) is lower for PH than Better, however importantly Better actually represents overlapping data with the PH dataset (Better consists of 2 proficient demonstrators). We use this as calibration between different dataset sizes, and we see that there is some effect of dataset size on our chosen metrics. Note that datatset size corresponds to number of episodes.

# 6 Discussion

Data curation is incredibly important for maximizing the potential of any given algorithm, and especially so in imitation learning. To curate data, one needs to establish a formalism for assessing data quality. The predominant notions of data quality today are solely centered around maximizing *state diversity* in a dataset, which ignore the quality of the demonstrated actions and how these factors interplay. In this work, we propose a holistic view of data curation centered around minimizing distribution shift. We demonstrate that action divergence and transition diversity are crucial factors in data quality that can often be controlled or measured, and we draw several valuable insights from this shedding light on effective strategies for curating demonstration datasets. We find that making actions more consistent tends to increase policy success. Furthermore, we observe a fundamental tradeoff between state diversity and action divergence — increasing state diversity can often come at the expense of action divergence. Instead of uniformly maximizing state diversity, we show that increasing transition diversity improves performance until the data coverage becomes too sparse. While the metrics presented in our experiments are often good measures of final performance, we find that comprehensively measuring data quality in practice for real world datasets can be quite challenging. Finally, we envision a broader set of metrics informed by our formalism for data quality that practitioners can use to curate datasets, and we believe our work is an important first step in formulating and evaluating these metrics.

**Limitations and Future Work:** We hope our analysis will inspire future research in data quality for imitation learning. Firstly, we hope to relax several theoretical assumptions like Gaussian noise in future work. The metrics defined in Section 5.2 do not necessarily capture the full scope of factors that comprise data quality. Future work should refine these metrics or find alternate ways of measuring data quality without requiring running the algorithm first, or potentially with only limited algorithm training. While we show the performance of several simple metrics on two interesting control settings, future work is needed in determining how best to use these metrics for guiding data collection. One option is to use the metrics as a score function that demonstrators might use as feedback. Another is to explicitly filter or weight data using the metrics. Furthermore, it would be beneficial to test the insights in this paper on real world settings, for example under more realistic models of system noise than simply Gaussian noise. Nonetheless, we believe our work to be an important first step in defining and measuring data quality.

**Fairness in robotics**: As discussed in our introduction, foundation models like GPT-3 trained on uncurated datasets have learned harmful biases and stereotypes from uncurated datasets. Efforts in the broader ML community have started to focus on curation as a means to solve this problem. In the policy learning setting in robotics, fairness and safety are less clearly defined. Our primary focus in this paper is on data curation for learning high performing policies, but we believe future work should find analogous notions of fairness in the robotics setting, and to consider how metrics of data quality that we define will affect this notion of fairness. For example, we might want to ensure data is collected over a diverse range of institutions rather than in just a select few locations. Our action consistency metric might artificially select only demonstrators from a small range of institutions, and so this should also be considered during data curation.

## Acknowledgments

This research was supported by ONR, DARPA YFA, Ford, and NSF Awards #1941722, #2218760, and #2125511. We also give special thanks to Stephen Tu for helping to streamline and improve several proofs for theorems in the text.

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

## A  Theoretical Results

: Given a policy $\pi_A$ and demonstrator $\pi_E$ and environment horizon length $H$, the distribution shift:

$$D_{\text{KL}}(\rho_{\pi_A}, \rho_{\pi_E}) \leq \frac{1}{H} \sum_{t=0}^{H-1} (H-t) \mathbb{E}_{s \sim \rho_{\pi_A}^t} \left[ D_{\text{KL}}(\pi_A(\cdot|s), \pi_E(\cdot|s)) \right]$$

*Proof.* Using the log-sum inequality:

$$D_{\text{KL}}(\rho_{\pi_A}^t, \rho_{\pi_E}^t) = \int_{s'} \rho_{\pi_A}^t(s') \log \frac{\rho_{\pi_A}^t(s')}{\rho_{\pi_E}^t(s')}$$

$$= \int_{s'} \left( \int_{s,a} \rho_{\pi_A}^{t-1}(s)\pi_A(a|s)\rho(s'|a,s) \right) \log \frac{\int_{s,a} \rho_{\pi_A}^{t-1}(s)\pi_A(a|s)\rho(s'|a,s)}{\int_{s,a} \rho_{\pi_E}^{t-1}(s)\pi_E(a|s)\rho(s'|a,s)}$$

$$\leq \int_{s'} \int_{s,a} \rho_{\pi_A}^{t-1}(s)\pi_A(a|s)\rho(s'|a,s) \log \frac{\rho_{\pi_A}^{t-1}(s)\pi_A(a|s)\rho(s'|a,s)}{\rho_{\pi_E}^{t-1}(s)\pi_E(a|s)\rho(s'|a,s)}$$

$$\leq \int_{s'} \int_{s,a} \rho_{\pi_A}^{t-1}(s)\pi_A(a|s)\rho(s'|a,s) (\log \frac{\rho_{\pi_A}^{t-1}(s)}{\rho_{\pi_E}^{t-1}(s)} + \log \frac{\pi_A(a|s)}{\pi_E(a|s)})$$

$$\leq \int_{s,a} \rho_{\pi_A}^{t-1}(s)\pi_A(a|s) (\log \frac{\rho_{\pi_A}^{t-1}(s)}{\rho_{\pi_E}^{t-1}(s)} + \log \frac{\pi_A(a|s)}{\pi_E(a|s)})$$

$$\leq \int_s \rho_{\pi_A}^{t-1}(s) \log \frac{\rho_{\pi_A}^{t-1}(s)}{\rho_{\pi_E}^{t-1}(s)} + \int_{s,a} \rho_{\pi_A}^{t-1}(s)\pi_A(a|s) \log \frac{\pi_A(a|s)}{\pi_E(a|s)}$$

$$\leq D_{\text{KL}}(\rho_{\pi_A}^{t-1}, \rho_{\pi_E}^{t-1}) + \mathbb{E}_{s \sim \rho_{\pi_A}^t} \left[ D_{\text{KL}}(\pi_A(\cdot|s), \pi_E(\cdot|s)) \right]$$

$$\leq D_{\text{KL}}(\rho^0(\cdot), \rho^0(\cdot)) + \sum_{j=0}^{t-1} \mathbb{E}_{s \sim \rho_{\pi_A}^j} \left[ D_{\text{KL}}(\pi_A(\cdot|s), \pi_E(\cdot|s)) \right] \quad \triangleright \text{Recursive}$$

$$\leq \sum_{j=0}^{t-1} \mathbb{E}_{s \sim \rho_{\pi_A}^j} \left[ D_{\text{KL}}(\pi_A(\cdot|s), \pi_E(\cdot|s)) \right]$$

$$D_{\text{KL}}(\rho_{\pi_A}, \rho_{\pi_E}) = \int_s \left( \frac{1}{H} \sum_{t=1}^{H} \rho_{\pi_A}^t(s) \right) \log \frac{\sum_{t=1}^{H} \rho_{\pi_A}^t(s)}{\sum_{t=1}^{H} \rho_{\pi_E}^t(s)}$$

$$\leq \int_s \sum_{t=1}^{H} \rho_{\pi_A}^t(s) \log \frac{\rho_{\pi_A}^t(s)}{\rho_{\pi_E}^t(s)}$$

$$\leq \frac{1}{H} \sum_{t=1}^{H} D_{\text{KL}}(\rho_{\pi_A}^t, \rho_{\pi_E}^t)$$

$$\leq \frac{1}{H} \sum_{t=1}^{H} \sum_{j=0}^{t-1} \mathbb{E}_{s \sim \rho_{\pi_A}^j} \left[ D_{\text{KL}}(\pi_A(\cdot|s), \pi_E(\cdot|s)) \right]$$

$$\leq \frac{1}{H} \sum_{t=0}^{H-1} (H-t) \mathbb{E}_{s \sim \rho_{\pi_A}^t} \left[ D_{\text{KL}}(\pi_A(\cdot|s), \pi_E(\cdot|s)) \right]$$

$$\square$$

Note that Ke et al. show a similar theorem, however they make the strong assumption of the f-divergence satisfying the triangle inequality, which is not true for the KL divergence we use in Theorem 4.1, and also yields a different bound. Furthermore, the implications of the divergence relationship on data quality (i.e. the data generating policy distribution) is not examined within this prior work. They are focusing more on the algorithmic perspective, as is common in prior work.

**Lemma 4.2**: Given learned policy $\pi_A$ and expert $\pi_E$, define assume that the policy is learned such that when $s \in \text{supp}(\rho^t_{\pi_E})$, $D_{\text{KL}}(\pi_A(\cdot|s), \pi_E(\cdot|s)) \leq \beta$. Then $\mathbb{E}_{s \sim \rho^t_{\pi_A}}[D_{\text{KL}}(\pi_A(\cdot|s), \pi_E(\cdot|s))] \leq \mathbb{E}_{s \in \rho^t_{\pi_A}}[\beta \mathbb{1}(s \in \text{supp}(\rho^t_{\pi_E})) + \mathbb{1}(s \notin \text{supp}(\rho^t_{\pi_E}))D_{\text{KL}}(\pi_A(\cdot|s), \pi_E(\cdot|s))]$

*Proof.* This follows by simple substitution:

$$
\begin{aligned}
\mathbb{E}_{s \sim \rho^t_{\pi_A}}[D_{\text{KL}}(\pi_A(\cdot|s), \pi_E(\cdot|s))] = \mathbb{E}_{s \in \rho^t_{\pi_A}}[&\mathbb{1}(s \in \text{supp}(\rho^t_{\pi_E}))D_{\text{KL}}(\pi_A(\cdot|s), \pi_E(\cdot|s)) \\
&+ \mathbb{1}(s \notin \text{supp}(\rho^t_{\pi_E}))D_{\text{KL}}(\pi_A(\cdot|s), \pi_E(\cdot|s))] \\
\leq \mathbb{E}_{s \in \rho^t_{\pi_A}}[&\beta \mathbb{1}(s \in \text{supp}(\rho^t_{\pi_E})) \\
&+ \mathbb{1}(s \notin \text{supp}(\rho^t_{\pi_E}))D_{\text{KL}}(\pi_A(\cdot|s), \pi_E(\cdot|s))]
\end{aligned}
$$

$\square$

**Theorem 4.3**: Given a policy $\pi_A$ and demonstrator $\pi_E$, assume that for state s, if $\rho^{t-1}_{\pi_E}(s) > 0$, then $\pi_A(a|s) = \pi_E(a|s)$. Assume that transitions are normally distributed with fixed and diagonal variance, $\rho(s'|s, a) = \mathcal{N}(\mu(s, a), \sigma^2 I)$, then the next state coverage probability is $P_S(s; N, \epsilon) \geq 1 - \left(1 - (\frac{c\epsilon}{\sigma})^d \exp\left(-\alpha^2 d\right)\right)^N - \exp\left(-(\alpha - 1)^2 d\right)$, where $d$ is the dimensionality of the state, $c$ is a constant, and $\alpha \geq 1$ is a parameter chosen to maximize the bound.

*Proof.* The proof follows by first discretizing the state space into length $\epsilon$ bins. Denote the probability mass of bin $b$ as $p_b$. First we note that for two independent samples from the same distribution $s'$ and $s'_{i,*}$, we can say that $P(|s' - s'_{i,*}|_\infty \geq \epsilon)$ is upper bounded by the probability that neither sample lands in the same bin. This is because of the fact that if $|s' - s'_{i,*}|_\infty \geq \epsilon$, then that implies the samples did not land in the same bin (if they did the infinity norm would be less than epsilon). Thus, we can also say that $P(\min_i |s' - s'_{i,*}|_\infty \geq \epsilon)$ is upper bounded by the probability that all of the samples $s'_{i,*}$ land in a different bin than $s'$. Thus:

$$
\begin{aligned}
P(\min_i |s' - s'_{i,*}|_\infty \geq \epsilon) &\leq P(\text{none lands in same bin}) \\
&\leq \sum_b p_b(1 - p_b)^N
\end{aligned}
$$

Next define a ball of radius $R$ around the mean $\mu(s, a)$ as $\text{Ball}(R)$. Using this ball we can partition the above inequality into two terms to yield an upper bound.

$$
\begin{aligned}
P(\min_i |s' - s'_{i,*}|_\infty \geq \epsilon) &\leq \sum_b p_b(1 - p_b)^N \\
&\leq \sum_{b \in \text{Ball}(R)} p_b(1 - p_b)^N + \sum_{b \notin \text{Ball}(R)} p_b(1 - p_b)^N
\end{aligned}
$$

We can upper bound everything inside the ball using the minimum Gaussian mass for the given radius $R$, or $p_b \geq (\frac{c\epsilon}{\sigma})^d \exp \frac{-R^2}{2\sigma^2}$ where $c = \frac{1}{\sqrt{2\pi}}$. For the second term, if $R$ is large enough, then we can assume $p_b$ is sufficiently small such that $(1 - p_b)^N \approx 1$.

$$
\begin{aligned}
P(\min_i |s' - s'_{i,*}|_\infty \geq \epsilon) &\leq \sum_{b \in \text{Ball}(R)} p_b \left(1 - \left(\frac{c\epsilon}{\sigma}\right)^d \exp \frac{-R^2}{2\sigma^2}\right)^N + \sum_{b \notin \text{Ball}(R)} p_b \\
&\leq \left(1 - \left(\frac{c\epsilon}{\sigma}\right)^d \exp\left(\frac{-R^2}{2\sigma^2}\right)\right)^N \left(\sum_{b \in \text{Ball}(R)} p_b\right) + \sum_{b \notin \text{Ball}(R)} p_b \\
&\leq \left(1 - \left(\frac{c\epsilon}{\sigma}\right)^d \exp\left(\frac{-R^2}{2\sigma^2}\right)\right)^N + \sum_{b \notin \text{Ball}(R)} p_b
\end{aligned}
$$

The second term is upper bounded by the probability that the $d$-dim Gaussian $s$ is farther in euclidean distance than $R$ from the mean, which can be written using the tail probability of the $\chi^2$ distribution.

$$P(\min_i |s' - s'_{i,*}|_\infty \geq \epsilon) \leq \left(1 - \left(\frac{c\epsilon}{\sigma}\right)^d \exp\left(\frac{-R^2}{2\sigma^2}\right)\right)^N + P(||s - \mu(s,a)||_2^2 \geq R^2))$$

$$\leq \left(1 - \left(\frac{c\epsilon}{\sigma}\right)^d \exp\left(\frac{-R^2}{2\sigma^2}\right)\right)^N + \left(1 - \chi^2(\frac{R^2}{\sigma^2})\right)$$

We know that for a $\chi^2$ random variable Y over $d$ sub Gaussians, $P(\frac{Y}{d} \geq (1+\delta)^2) \leq \exp\left(\frac{-d\delta^2}{2}\right)$ for $\delta \geq 0$ [52]. Thus assuming $R \geq \sigma\sqrt{d}$, we can write:

$$P(\min_i |s' - s'_{i,*}|_\infty \geq \epsilon) \leq \left(1 - \left(\frac{c\epsilon}{\sigma}\right)^d \exp\frac{-R^2}{2\sigma}\right)^N + \exp\left(-\left(\frac{R}{\sigma} - \sqrt{d}\right)^2\right)$$

Now, for any $\alpha = \frac{R}{\sigma\sqrt{d}} \geq 1$, we can rewrite the above bound as:

$$P(\min_i |s' - s'_{i,*}|_\infty \geq \epsilon) \leq \left(1 - \left(\frac{c\epsilon}{\sigma}\right)^d \exp\left(-\alpha^2 d\right)\right)^N + \exp\left(-(\alpha-1)^2 d\right)$$

Starting from Definition 1:

$$P_S(s; N, \epsilon) = P(\min_i ||s' - s'_{*,i}||_\infty \leq \epsilon)$$

$$= 1 - P(\min_i |s' - s'_{i,*}|_\infty \geq \epsilon)$$

$$\geq 1 - \left(1 - \left(\frac{c\epsilon}{\sigma}\right)^d \exp\left(-\alpha^2 d\right)\right)^N - \exp\left(-(\alpha-1)^2 d\right)$$

$\square$

## A.1   Generalization under System Noise

**Definition 2.** *Given a policy $\pi_A$, a data generating policy $\pi_E$ and a starting state $s$, define the probability of next state coverage $P_B(s, \mu; N) = 1 - \cap_i P(||s'_i - \mu||^2 \geq ||s'_{i,*} - \mu||^2)$, where $s'_{i,*} \sim \rho^t_{\pi_E}(\cdot|s)$ are next state samples from the expert starting at $s$, and $s'_i \sim \rho^t_{\pi_A}(\cdot|s)$ are next state samples from $\pi_A$ starting at $s$.*

Intuitively $P_B(s; N)$ is the probability that given $N$ chances, the hyper-sphere defined by the L2 distance from $\mu$ to a sampled data point contains a sample from the learned policy. Here the hyper-sphere represents the set of next states that the policy can generalize to. The $N$ chances approximate the effect of having more data samples to leverage for generalization. For policies learned with neural networks, this aims to represent the "interpolation" capacity of these models among the training samples.

**Theorem A.1.** *Given a policy $\pi_A$ and deterministic demonstrator $\pi_E$, assume that for state $s$, if $\rho^{t-1}_{\pi_E}(s) > 0$, then $\pi_A(a|s) = \mathcal{N}(\pi_E(s), \sigma^2_{\pi}I)$. Assume that transitions are normally distributed with fixed and diagonal variance, $\rho(s'|s,a) = \mathcal{N}(\mu(s,a), \sigma^2 I)$, where $\mu(s,a) = s + \alpha a$ are simplified linear dynamics for scalar $\alpha \in \mathbb{R}$. Then the next state coverage probability is $P_B(s, \mu(s,a); N) = 1 - \left(1 - F_d\left(\frac{1}{1+\alpha^2\left(\frac{\sigma_p}{\sigma}\right)^2}\right)\right)^N$, where $F_d(x)$ is the CDF of the f-distribution of dimension d.*

*Proof.* First we note that both $\Delta_i = s'_i - \mu(s,a)$ and $\Delta_{i,*} = s'_{i,*} - \mu(s,a)$ are zero mean, and the events for $\Delta_i \geq \Delta_{i,*}$ for all $i$ are independent. Thus:

$$P_B(s, \mu; N) = 1 - \cap_i P(||\Delta_i||^2 \geq ||\Delta_{i,*}||^2)$$

$$= 1 - \left(1 - P(||\Delta_i||^2 \leq ||\Delta_{i,*}||^2)\right)^N$$

$$= 1 - \left(1 - P\left(\frac{||\Delta_i||^2}{||\Delta_{i,*}||^2} \leq 1\right)\right)^N$$

Note that $||\Delta_i||^2$ is just a $\chi_d^2$ random variable with $d$ degrees of freedom, and likewise for $||\Delta_{i,*}||^2$. The former has variance $\sigma^2 + (\alpha\sigma_p)^2$ (transition plus learned policy noise), while the latter has variance $\sigma^2$ (just transition noise). The f-distribution is defined for two $\chi_d^2$ variables $X$ and $Y$ of dimension $d$ as the distribution of $Z = \frac{X/d}{Y/d}$. Thus with a change of variables to $X$ and $Y$, and denoting $F_d$ as the CDF of the f-distribution of dimension $d$:

$$P\left(\frac{||\Delta_i||^2}{||\Delta_{i,*}||^2} \leq 1\right) = P\left(\frac{(\sigma^2 + (\alpha\sigma_p)^2)X}{\sigma^2 Y} \leq 1\right)$$

$$= P\left(\frac{X}{Y} \leq \frac{\sigma^2}{\sigma^2 + (\alpha\sigma_p)^2}\right)$$

$$= F_d\left(\frac{\sigma^2}{\sigma^2 + (\alpha\sigma_p)^2}\right)$$

$$= F_d\left(\frac{1}{1 + \alpha^2 \left(\frac{\sigma_p}{\sigma}\right)^2}\right)$$

Plugging the above expression into the expression for $P_B$:

$$P_B(s, \mu; N) = 1 - \left(1 - F_d\left(\frac{1}{1 + \alpha^2 \left(\frac{\sigma_p}{\sigma}\right)^2}\right)\right)^N$$

$\square$

In Theorem A.1, while the probability of next state coverage is not immediately interpretable, we can still intuitively recognize that high $\sigma_s$ can improve the coverage likelihood of the learned policy even under notable $\sigma_p$, and as $N$ gets bigger even less $\sigma_s$ is needed, despite the fact that system noise is present even under the learned policy. Intuitively, we once again see that increasing $N$ has a significant effect on the coverage likelihood. In terms of noise, what matters is the ratio of policy to system noise, where increasing this ratio leads to sharp drops in performance at some cutoff based on $N$. We visualize this coverage probability in Fig. 4 under increasing ratios of policy to system noise for different values of $N$.

## B   Metrics of Data Quality

Having formalized action divergence and transition diversity in Sec. 4 as two fundamental considerations in a dataset, how can we *measure* these properties in a given dataset?

**Action Variance**: To measure action consistency, the empirical form of the objective in Eqn. 7 is intractable without access to the underlying expert action distribution $\pi_E$. Instead we propose using the empirical variance of the action distribution in the data to approximate the "spread" of the data. In continuous state spaces, we can estimate variance using a coverage distance $\epsilon$ to cluster nearby states, and then measuring the per dimension variance across the corresponding actions within said cluster. Defining a cluster to be $C(s, \mathcal{D}) = \{\tilde{s}, \tilde{a}, \tilde{s}' \in \mathcal{D} : ||s - \tilde{s}|| \leq \epsilon\}$, we can compute the variance as:

$$\text{ActionVariance}(\mathcal{D}) = \frac{1}{|D|} \sum_{s,a \in \mathcal{D}} (a - \sum_{\tilde{s}, \tilde{a}, \tilde{s}' \in C(s, \mathcal{D})} \tilde{a})^2 \tag{9}$$

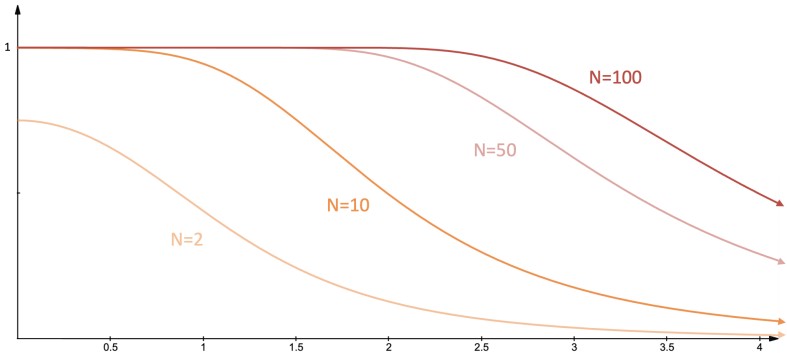

Figure 4: $P_B(s, \mu; N)$ (y-axis) from Theorem A.1 plotted for four-dimensional state under $N \in [2, 10, 50, 100]$, but varying the ratio of policy to system noise (x-axis is $\frac{\sigma_p}{\sigma}$). We see that under this more loose coverage model, with lots of samples, adding system noise can make coverage likely even under double or triple the noise in the learned policy.

The choice in $\epsilon$ corresponds to the *generalization* of the learning model to nearby states, similar to the notion of coverage in Definition 1. We use this metric of action consistency in Sec. 5 to study human generated datasets of various quality.

**State Similarity**: To measure the consistency of states, we approximate the number of "nearby" states using the same clustering process as in the Action Variance metric, and measure the expected cluster size as a fraction of the overall data.

$$\text{StateSimilarity}(\mathcal{D}) = \frac{1}{|D|} \sum_{s,a \in \mathcal{D}} |C(s, \mathcal{D})| \tag{10}$$

While these approximate forms do not encapsulate the full spectrum of possible metrics, we believe these metrics help advance our empirical understanding of data quality for imitation learning. In section 5.2 in the main text, we analyze these metrics of data quality in several environments across different dataset sources.

## C  Results

The performance results under system noise, policy noise, and both noises are shown with a broader sweep for both *PMObstacle* and *Square* in the tables below.

|  | $\sigma_s = 0.01$ | $\sigma_s = 0.02$ | $\sigma_s = 0.03$ | $\sigma_s = 0.04$ | $\sigma_s = 0.01$ | $\sigma_s = 0.02$ | $\sigma_s = 0.03$ | $\sigma_s = 0.04$ |
|---|---|---|---|---|---|---|---|---|
| SCRIPTED | 100 | 100 | 99 | 96 | | | | |
| $\sigma_s = 0.01$ | 97.7(1.5) | 95.7(0.7) | 96.7(1.1) | 93.3(0.7) | 90.3(7.1) | 90.0(8.2) | 94.0(2.9) | 87.7(3.7) |
| $\sigma_s = 0.02$ | 98.7(0.5) | 98.0(0.5) | 97.7(1.0) | 92.7(1.2) | 99.7(0.3) | 98.0(0.9) | 94.3(1.4) | 92.3(2.0) |
| $\sigma_s = 0.03$ | 98.3(0.7) | 98.0(0.8) | 99.0(0.5) | 95.0(0.9) | 99.7(0.3) | 98.7(0.5) | 97.7(0.5) | 95.7(1.1) |
| $\sigma_s = 0.04$ | 100.0(0.0) | 100.0(0.0) | 99.3(0.3) | 96.7(0.7) | 100.0(0.0) | 99.0(0.5) | 98.7(0.5) | 96.7(1.4) |

Table 2: **System Noise**: Success rates (and standard error) for BC in PMObstacle, for 1000 episodes (left) and 10 episodes (right) of data, under system noise. Rows correspond to injecting gaussian system noise ($\sigma_s$) into the *dataset* of increasing variance, and columns correspond to injecting noise during *evaluation*. The diagonal in both sub-tables represents evaluating in distribution. **Left**: For large datasets, higher system noise during evaluation tends to decrease the performance of each model (rows left to right), but more system noise during training generally produces the best models (columns top to bottom). **Right**: For small datasets, we observe a similar but exaggerated effect as the left table.

|  | $\sigma_s = 0.01$ | $\sigma_s = 0.02$ | $\sigma_s = 0.03$ | $\sigma_s = 0.04$ | $\sigma_s = 0.01$ | $\sigma_s = 0.02$ | $\sigma_s = 0.03$ | $\sigma_s = 0.04$ |
|---|---|---|---|---|---|---|---|---|
| SCRIPTED | 100 | 100 | 99 | 96 | | | | |
| $\sigma_p = 0.01$ | 94.0(1.7) | 94.0(2.4) | 94.7(1.8) | 91.3(1.4) | 78.0(8.6) | 78.7(6.7) | 81.3(5.0) | 81.3(5.8) |
| $\sigma_p = 0.02$ | 87.7(2.0) | 92.3(2.0) | 90.7(2.0) | 91.3(2.2) | 88.0(9.4) | 78.7(4.4) | 80.7(3.1) | 80.3(3.2) |
| $\sigma_p = 0.03$ | 97.0(0.9) | 99.0(0.5) | 97.0(0.0) | 95.0(0.8) | 88.7(3.2) | 82.7(5.4) | 88.7(5.2) | 85.7(4.4) |
| $\sigma_p = 0.04$ | 86.7(4.3) | 91.0(2.4) | 93.3(1.4) | 92.7(1.5) | 88.3(6.5) | 88.0(4.5) | 86.0(5.9) | 82.3(2.0) |

Table 3: **Policy Noise**: Success rates (and standard error) for BC in PMObstacle, for 1000 episodes (left) and 10 episodes (right) of data, under learned policy noise. Rows correspond to injecting gaussian policy noise ($\sigma_p$) into the *expert* of increasing variance, and columns correspond to injecting system noise during *evaluation*. **Left**: For large datasets, unlike system noise in Table 2, more policy noise during training often produces the worst models (columns top to bottom). **Right**: For small datasets, adding policy noise produces large variance in performance across runs. Importantly, the datasets in each row have the same observed state diversity as the corresponding row in Table 2, but performance is almost universally lower in both sub-tables here, supporting the idea that state diversity is a coarse metric for success.

|  | $\sigma_s = 0.01$ | $\sigma_s = 0.02$ | $\sigma_s = 0.03$ | $\sigma_s = 0.04$ | $\sigma_s = 0.01$ | $\sigma_s = 0.02$ | $\sigma_s = 0.03$ | $\sigma_s = 0.04$ |
|---|---|---|---|---|---|---|---|---|
| SCRIPTED | 100 | 100 | 99 | 96 | | | | |
| $\sigma_p = 0.01$ | 96.3(2.2) | 99.3(0.5) | 97.7(0.3) | 93.3(1.0) | 99.7(0.3) | 98.0(1.2) | 96.7(0.5) | 96.3(1.4) |
| $\sigma_p = 0.02$ | 98.0(0.5) | 98.3(0.5) | 97.7(0.7) | 94.7(1.0) | 99.3(0.5) | 98.7(1.1) | 96.0(2.2) | 94.7(1.7) |
| $\sigma_p = 0.03$ | 98.0(0.8) | 96.7(1.0) | 98.3(1.0) | 96.3(0.7) | 95.0(2.1) | 97.7(0.5) | 97.7(0.5) | 93.3(2.2) |
| $\sigma_p = 0.04$ | 98.7(0.5) | 99.0(0.5) | 97.3(0.3) | 95.0(0.8) | 100.0(0.0) | 99.7(0.3) | 99.0(0.8) | 93.7(3.1) |

Table 4: **System Noise + Policy Noise**: Success rates (and standard error) for BC in PMObstacle, for 1000 episodes (left) and 10 episodes (right) of data, under learned policy noise for a fixed amount of system noise ($\sigma_p = 0.03$). Here we see how system noise improves the robustness of the model to added policy noise.

|  | $\sigma_s = 0.05$ | $\sigma_s = 0.1$ | $\sigma_s = 0.2$ | $\sigma_s = 0.3$ | $\sigma_s = 0.4$ |
|---|---|---|---|---|---|
| $\sigma_s = 0.05$ | 55.7(5.3) | 50.0(5.4) | 27.0(3.9) | 12.0(2.4) | 7.3(2.2) |
| $\sigma_s = 0.1$ | 69.7(5.5) | 69.0(3.7) | 57.3(6.0) | 50.3(6.0) | 22.7(3.5) |
| $\sigma_s = 0.2$ | 67.7(9.6) | 68.7(12.6) | 82.0(3.4) | 74.3(2.7) | 50.3(1.7) |
| $\sigma_s = 0.3$ | 47.0(4.9) | 53.3(5.9) | 54.3(3.0) | 50.7(4.3) | 38.7(7.3) |
| $\sigma_s = 0.4$ | 31.3(5.0) | 37.7(8.2) | 48.3(9.7) | 49.0(8.7) | 44.0(5.3) |

Table 5: **System Noise, 200ep**: Success rates for BC in Square, for 200 episodes of data, under system noise. Rows correspond to injecting gaussian system noise ($\sigma_s$) into the *dataset* of increasing variance, and columns correspond to injecting noise during *evaluation*. The diagonal in both sub-tables represents evaluating in distribution. In both sub-tables we see how policies with low data coverage (low system noise) generalize the worst to increasing noise at test time. More system noise during training generally produces the best models (columns top to bottom).

|  | $\sigma_s = 0.05$ | $\sigma_s = 0.1$ | $\sigma_s = 0.2$ | $\sigma_s = 0.3$ | $\sigma_s = 0.4$ |
|---|---|---|---|---|---|
| $\sigma_s = 0.05$ | 40.0(1.2) | 33.7(3.1) | 16.7(1.4) | 4.3(1.0) | 2.0(0.5) |
| $\sigma_s = 0.1$ | 42.3(4.0) | 39.7(3.8) | 31.3(3.7) | 19.7(3.1) | 10.0(1.6) |
| $\sigma_s = 0.2$ | 70.0(7.0) | 73.7(5.4) | 69.7(0.7) | 55.3(1.2) | 27.0(2.2) |
| $\sigma_s = 0.3$ | 57.3(3.1) | 58.7(3.1) | 64.7(1.4) | 60.7(0.3) | 44.7(2.2) |
| $\sigma_s = 0.4$ | 30.0(7.0) | 33.7(6.4) | 39.7(5.7) | 39.7(6.5) | 36.7(6.9) |

Table 6: **System Noise, 50ep**: Success rates for BC in Square, for 50 episodes of data, under system noise. Rows correspond to injecting gaussian system noise ($\sigma_s$) into the *dataset* of increasing variance, and columns correspond to injecting noise during *evaluation*. The diagonal in both sub-tables represents evaluating in distribution. In both sub-tables we see how policies with low data coverage (low system noise) generalize the worst to increasing noise at test time. More system noise during training generally produces the best models (columns top to bottom).

|  | $\sigma_s = 0.05$ | $\sigma_s = 0.1$ | $\sigma_s = 0.2$ | $\sigma_s = 0.3$ | $\sigma_s = 0.4$ |
|---|---|---|---|---|---|
| $\sigma_p = 0.005$ | 69.0(3.7) | 59.0(3.7) | 34.0(1.7) | 20.7(2.4) | 7.0(1.6) |
| $\sigma_p = 0.01$ | 80.7(3.7) | 78.0(4.2) | 57.7(2.4) | 38.0(1.7) | 23.0(2.2) |
| $\sigma_p = 0.02$ | 62.3(7.8) | 71.7(6.1) | 73.0(3.9) | 65.3(2.8) | 43.3(3.6) |

Table 7: **Policy Noise, 200ep**: Success rates for BC in Square, for 200 episodes of data, under learned policy noise. Rows correspond to injecting gaussian policy noise ($\sigma_p$) into the *dataset* of increasing variance, and columns correspond to injecting noise during *evaluation*. In the high data regime, we see that more policy noise tends to improve performance (columns top to bottom), since the noise is unbiased so with enough samples from the scripted policy, the model will recover an unbiased policy.

|  | $\sigma_s = 0.05$ | $\sigma_s = 0.1$ | $\sigma_s = 0.2$ | $\sigma_s = 0.3$ | $\sigma_s = 0.4$ |
|---|---|---|---|---|---|
| $\sigma_p = 0.005$ | 32.7(3.8) | 30.3(3.2) | 18.0(3.6) | 7.0(0.8) | 5.7(1.2) |
| $\sigma_p = 0.01$ | 61.3(4.3) | 59.0(6.7) | 48.3(3.7) | 29.7(1.7) | 19.3(1.4) |
| $\sigma_p = 0.02$ | 57.7(3.2) | 58.3(3.1) | 49.3(0.5) | 41.3(0.5) | 28.3(2.4) |

Table 8: **Policy Noise, 50ep**: Success rates for BC in Square, for 50 episodes of data, under learned policy noise. Rows correspond to injecting gaussian policy noise ($\sigma_p$) into the *dataset* of increasing variance, and columns correspond to injecting noise during *evaluation*. As the amount of data is reduced, there is a significant drop in performance for added policy noise in the dataset, along with higher performance variation compared to 200eps, since the policy can no longer recover an unbiased policy.

