# OpenReview forum: "Data Quality in Imitation Learning"
_NeurIPS.cc/2023/Conference — NeurIPS 2023 poster_

### Official Review · Reviewer_AXdc · 2023-06-30

**Soundness:** 2 fair
**Presentation:** 2 fair
**Contribution:** 1 poor
**Rating:** 2
**Confidence:** 5

**Summary:**

The paper analyze several theoretical property of the expert demonstration distribution to the performance of the learnt policy in imitation learning. The paper claims that people should provide consistent and transition and state diversity.

**Strengths:**

The paper defines several quantity to clear express the meaning of the properties. The theorems are sound to me.

**Weaknesses:**

The results claimed by the paper are straightforward to me, which makes the contribution less significant. The theoretical results are basically quick derivation of standard machinery of MDP analysis, which are trivial and straightforward.

The paper claims that expert should provide consistent demo based on BC analysis. However, the paper does provide experiment of the types of imitation learning methods like transformer architecture to handle multi-modal distribution.


**Questions:**

See weakness.

**Limitations:**

See weakness.

---

> ### Author Rebuttal · Authors · 2023-08-10
>
> We thank the reviewer for their thoughtful review.
>
> **Straightforward results**
>
> We agree with the reviewer that several of the conclusions are intuitive; however, we disagree that they are trivial or straightforward. See our main response comment (Straightforward Results).
>
> The reviewer also states that the theoretical results are based on simple derivations and the “standard machinery of MDP analysis”, but simplicity is not the same as significance. The impact of theoretical results depends on how well they explain practical results, not how simple the proofs are to understand. Our theoretical claims are actually supported by the practical results on several environments, and as exemplified above this could have high significance in the field of imitation learning.
>
> On the point of novelty, we emphasize that ours is the first work to form a full picture of data quality and glean useful properties from it. We are hoping this work inspires a paradigm shift to focus on improving data quality rather than just changes to any given algorithm.
>
> **Transformer Architectures for Multimodality**
>
> The reviewer requests additional experiments involving transformer based architectures that can handle multimodality. We have included these results for PointMassObstacle in the attached one-pager. Overall, we see that transformers still suffer from policy noise (i.e., action divergence), while they benefit from system noise (i.e., transition diversity).
>
> Recently, prior work has studied these transformer architectures in the multi-task setting as well, for example RT-1 [1] and RT-2 [2]. They use massive datasets and incredibly large transformer architectures, for relatively simple tasks (pick and place), and still their performance leaves much to be desired. This is further evidence that large transformer-based architectures can’t overcome challenges associated with uncurated datasets, even as datasets grow to hundreds of thousands or millions of trajectories. As the field trends towards bigger models and datasets, we argue intelligent data curation (e.g. through metrics of data quality like those proposed in Section 5.2) might be a better use of time and resources.
>
> Please let us know if you have any further questions or concerns, and in light of our response, we hope that you will consider raising your score.
>
> [1] Brohan, Anthony, et al. "Rt-1: Robotics transformer for real-world control at scale." arXiv preprint arXiv:2212.06817 (2022).
>
> [2] Brohan, Anthony, et al. "RT-2: Vision-Language-Action Models Transfer Web Knowledge to Robotic Control." https://robotics-transformer2.github.io/assets/rt2.pdf

---

### Official Review · Reviewer_WXvd · 2023-07-06

**Soundness:** 3 good
**Presentation:** 3 good
**Contribution:** 3 good
**Rating:** 6
**Confidence:** 3

**Summary:**

This paper considers how to evaluate  the quality of a given dataset for an imitation learning task.  It is well-known that IL suffers from distribution shift. While much work on IL focuses on improving algorithms, the authors propose that another important approach to making IL more effective is to curate better datasets and demonstrations from experts. Existing approaches to evaluating the quality of datasets focus on state coverage / diversity, trying to ensure that the expert demonstration contains instances of a variety of different states.
The authors discuss in some depth how state diversity is an incomplete measure of dataset quality, specifically because it ignores the actions taken in the dataset; the goal of the paper is to find an improved metric for evaluating dataset quality. he paper proposes action diversity and transition diversity.  The authors then consider  the distribution shift from the expert policy to the learned policy under a given f-divergence (KL divergence by default), as having a large distribution shift can cause problems in IL.
They then show a number of properties of their measure and relate it to action diversity and transition diversity , and discuss how these metrics can be applied to improve IL in general (4.3, Implications for Data Curation). They claim that their work suggests that action consistency in the expert demonstrations leads to better outcomes, while state diversty and system noise in the dataset are good or bad depending on context.
Finally, they perform experiments on two simple simulationtasks, comparing the effects of adding policy noise to adding system noise in the training set and comparing how demonstration datasets of varying (known) quality correlate with various metrics.

**Strengths:**

The main idea of finding improved metrics for dataset quality, above and beyond state space coverage, is interesting. Furthermore, the dichotomy that their analysis exposes between expanding state space coverage via higher action divergence versus higher transition divergence is an interesting one, showing that not all state space coverage is created equal. The theoretical results linking Q with action divergence are also interesting and well-presented.
While there are questions left by the experiments (see Weaknesses and Questions) one extremely interesting finding is that the additon of a small amount of system noise vastly improves the effectiveness of training with policy noise (compare Figure 2 (c)(d) with Figure 2 (i)(j)), especially when the number of demos is low. The effect of this is so strong that it would have been better to focus more on it when discussing the results.

With a few minor exceptions, the paper is also very well written. The essential ideas are clearly stated and easy for the reader to absorb.

**Weaknesses:**

This paper would be greatly improved by reducing repetition and adding more experiments.  The points made in the Intro are repeated at least 5 times.  The paper would be much better if the authors agree to remove this redundancy and add more thorough experiments.

he biggest weakness is a key property of the proposed measure (equation (6)), which is tied to a strength of the work ("data quality is heavily tied to the algorithm A that leads to the learned policy \pi_A as well as the expert policy \pi_E"): since it takes the learned policy \pi_A as well as the expert policy \pi_E, actually measuring the quality Q may require running A, which somewhat diminishes the value of the data quality metric since the it may not be usable when collecting the data (note that most common metric, state diversity in the training dataset, can be measured without running A). Even one of the proposed proxies for Q, action divergence, requires having the learned policy to measure against the expert policy.
There is an inconsistency in lines 196 and 199-200 with Theorem 4.1: it says the data quality is "lower bounded by the action divergence" but Theorem 4.1 shows that action divergence provides an upper bound to the data quality. Since upper and lower bounds for dataset quality have different implications this is very important to fix.
While Figure 2 (i)(j) represents the most interesting finding (in my opinion) of this work, it doesn't escape notice that there is no corresponding Figure 2 (k)(l) showing the same plots for the Square nut task. These should absolutely be included to show whether or not the finding in (i)(j) generalizes to at least one other task (otherwise there is the possibility that (i)(j) is simply a property of one particular setting).
The authors do not include a limitations section.
Minor weaknesses and typos:
In (3), the sum should go for t=1 to T, not H
Figure 2 caption: In "X-axis corresponds…" should probably write "injected system noise" twice (rather than "injected Gaussian noise" the first time and "injected system noise" the second time). They're both system noise and both Gaussian, right? Writing it differently may convey the mistaken impression that they might not be both Gaussian, whereas the difference is just when the noise is added. Notation is also confusing, it seems \sigma_s is used both to refer to system noise at evaluation (in the figure itself) and to system noise added in the training set (line 342).
Table 1: instead of merely stating that state similarty is the "opposite of state diversity" (line 360), it would be much better to have a more precise definition. Additionally, it is claimed that "state diversity once again is not correlated with performance on the task" (line 370, just above Table 1) but Better/Okay/Worse all seem to have similar state similarity values while Proficient Human (PH) has lower state similarity (i.e. higher state diversity) and has significantly better success rates – doesn't this mean that it is correlated? Claims of no correlation should probably come with the actual correlation number, since it isn't obvious from the table that there really is no correlation.
Equation (6) should probably be in a Definition since it's one of the key contributions of the work.
4.3 Implications for Data Curation might be better placed after the experiments section.

**Questions:**

This paper would be greatly improved by reducing repetition and adding more experiments.  The points made in the Intro are repeated at least 5 times.  The paper would be much better if the authors agree to remove this redundancy and add more thorough experiments.

A number of the authors' conclusions (regarding what their results imply about creating and evaluating IL datasets) do not seem to follow obviously from their results and experiments. For instance, they claim in 4.3 (line 294-295): "we posit that expert demonstrators should encourage paths with high system entropy [up to a point]" – this doesn't really seem to follow from their experiments, in which system noise is added to a fixed policy, rather than changing the policy itself.
The authors claim to show that action divergence is undesirable and leads to worse outcomes, while transition diversity (in moderation) can improve results; this is demonstrated by adding policy noise during training and comparing the resulting policy to that learned from a dataset with system noise. However, this only shows that action divergence is harmful when it results from noise being artificially added (in which case the expert can be viewed as only partly trying to achieve the task, while also being partly random).
What about the effect of action divergence resulting from multi-modality (i.e. when many very different courses of action can achieve the goal in a reasonable way)? It is not at all clear that the proposed measures of data quality will work for multi-modal tasks –  which have different solutions, e.g. even simple ones like moving around an obstacle (where you can go to the left or right).
It would be interesting to consider how this can be used to guide human expert demonstrations; while I'm not convinced by the discussion in section 4.3, at least without more details (see the first item under Questions), there are possibly other interesting avenues to consider, such as adding noise to the human expert's controls when conducting demonstrations.

**Limitations:**

There is no Limitations section in the paper.

The limitations of the work are only very briefly addressed (lines 390-392) and need more expanding; in particular, the weakness mentioned (that the data quality metric proposed is very hard to measure in real datasets) is a serious one and deserves a much broader discussion.

---

> ### Author Rebuttal · Authors · 2023-08-10
>
> We thank the reviewer for their thoughtful review, and we are glad the reviewer found both the theoretical and empirical analysis important and interesting.
>
> **Reducing repetition**
>
> We are sorry that the reviewer felt the paper was repetitive! Our intention was to clearly state the takeaways, but we recognize that the text can be shortened. We will fix this in the main text.
>
> **How is data quality (Eq.6) useful if we need to first run the algorithm to evaluate it?**
>
> See our main comment response (same title).
>
> **Theorem 4.1: Lower bound**
>
> The reviewer is correct that the Theorem 4.1 shows an upper bound for the KL divergence, however *quality* is the negative of this KL divergence, and is thus lower bounded.
>
> **Adding Figure 2(k)(l)**
>
> We agree that showing effects of combined system and policy noise for the Square task is important, so we have included findings for this figure in the one pager (see main response), and will add this to the main text. To summarize the results, we once again find that adding system noise to the training data makes the learned policy more robust to policy noise. We thank the reviewer for pointing this out and hope this completes the story of our experimental section.
>
> **Adding a Limitations Section**
>
> We apologize for not explicitly including our limitations as part of the text. We will address our limitations in the updated text, and we have included a draft of the limitations in the attached one pager.
>
> **Experimental evidence for high system entropy up to a point**
>
> The claim that we should encourage paths with high system entropy up to a point is backed up by results for the Square dataset (a more complex task than PointMass Obstacle which has a much larger state space), where increasing system noise only helps performance up to a point, for both 200 and 50 episode datasets.
>
> **Action divergence resulting from multi-modality**
>
> The reviewer raises the point that action divergence resulting from multi-modality is different from action divergence stemming from randomness. We highlight that in the PointMassObstacle task, there is in fact multi-modality already present, where the data consists of going either left or right around the obstacle. Without this multi-modality, we find that performance increases to near 100% even for many values of both system and policy noise. This emphasizes the point that action divergence resulting from multi-modality is equally or more harmful than action divergence stemming from randomness. We will add this result to the Appendix and clarify this in the main text. We acknowledge that better metrics are needed for measuring this multi-modality, and we are excited to pursue this in future research.
>
> We also evaluate PointMassObstacle with transformer models that can supposedly handle more multimodality, however as seen in the results (one-pager), even transformers suffer from policy noise but benefit from system noise. This provides additional evidence of how existing models struggle to learn from multi-modal and noisy demonstrations.
>
> **State similarity for PH vs. Better/Okay/Worse**
>
> The reviewer is correct to note that state similarity is lower for PH than Better, however importantly Better actually represents overlapping data with the PH dataset (Better consists of 2 proficient demonstrators). We use this as calibration between different dataset sizes, and we see that there is some effect of dataset size on our chosen metrics. Furthermore, within Better / Okay / Worse we see either the opposite trend (Square) or uncorrelated performance (Can) with state similarity, and thus it would not be fair to say more state diversity leads to better policies. We have included this point in the analysis.
>
> **Miscellaneous**
>
> - We thank the reviewer for the various typos pointed out, and will fix them for the final manuscript.
> - We apologize that notation was confusing for using \sigma_s as both training and evaluation noise. We have now distinguished these two in the paper and figure.
> - A precise definition for state similarity is present in Appendix B.
>
> Please let us know if you have any further questions or concerns, and in light of our response, we hope that you will consider raising your score.

---

> > ### Comment · Reviewer_WXvd · 2023-08-16
> > **Thanks to Authors for their response to my review.**
> >
> > I appreciate the Author responses to my review and have also read the other reviews and responses.  I appreciate that the Authors have added a Limitations section.  After careful consideration I maintain my rating of Weak Accept, but I do feel that Data Quality is an important topic and that the authors should be encouraged to continue working on this topic!

---

### Official Review · Reviewer_9hHD · 2023-07-09

**Soundness:** 3 good
**Presentation:** 4 excellent
**Contribution:** 3 good
**Rating:** 6
**Confidence:** 2

**Summary:**

This paper studies imitation learning (IL) from a theoretical perspective. It proposes the idea that we should curate the data to improve performance, rather than solely relying on designing more robust algorithms, which is a more common practice in the field. The authors propose data quality for a particular IL algorithm, which is the negative of the divergence between the expert policy and the learned policy using that algorithm. They then show theoretically how two properties, action divergence and transition diversity, are linked to data quality. The authors then study the finite-data regime under a very simplistic setting, and finally show some results of studying their metrics on PMObstacle, Square, and Robomimic domains.

**Strengths:**

The presentation / quality of the writing is top-notch. I really appreciated the clear descriptions throughout the paper, starting from the well-written introduction to the explicit problem setting to the way the authors use English to describe and contextualize all the math.

Related work section is comprehensive. I like how it addresses both algorithmic and data robustness.

I also really like the framing of Equation 6. The insight that data quality is a function of the learning algorithm is very nice.

**Weaknesses:**

I am unfortunately not well-versed in IL theory, so I cannot comment on the novelty of the work with respect to the rest of the field. Therefore, I will only evaluate the paper on the merits of the contributions stated by the authors.

My only major weakness is about the unrealistic conditions underlying some of the theory.
* One of the conditions of Lemma 4.2 is that states in the support of the expert policy would have low divergence between actions under the expert policy and the learned policy. The authors say in L226 that with a good learning algorithm, this bounding constant \beta would be small, but I don't see why this would be true in general. For BC, maybe it makes sense, but for other IL algorithms, a "good" algorithm could sacrifice exactly matching the expert at some states, for the sake of getting better interpolation/generalization to states outside of the support of the expert policy.
* Theorem 4.3 studies only a very simple setting where the environment dynamics are fixed Gaussian with diagonal variance. I'm not sure how much we can take away from this Theorem as this condition will almost never hold in practice. As a consequence, I find takeaways like that in L254 to be misleading because one cannot draw general conclusions from such a restricted Theorem.

Other comments:
* One more broad concern I have is that most of the robotics domains studied in the RL literature have deterministic dynamics, including Robomimic (ref: https://arxiv.org/abs/2305.14550) and Mujoco. So how can we obtain transition diversity in such settings?
* Theorem 4.1 only makes sense with stochastic policies due to the KL divergence. It's not clear how to apply these findings to deterministic policies, which are quite common (and in fact, are used in the experiments of this paper).
* L230, L290: It's unclear what the authors mean when they say the data should maximize system noise. That's a property of the environment dynamics, not the expert policy; we don't have control over it!
* It is hard to gain too many insights from Table 1. A simpler explanation for the observed success rates is that as task horizon increases while data size stays fixed, BC simply has less data for each state, leading to worse policies. I'm not really sure why action variance and transition diversity come into play here.

Minor things:
* The authors use the word "expert" throughout the paper. I would rather say the "behavior policy" or "data generation policy", because the whole point of this paper is that we are trying to curate the dataset, e.g. by introducing more diversity, which means the data may not end up looking like what an expert would do.
* It would be good to say in the preliminaries that you're in a continuous state/action space setting.
* L73: remove "how"
* L167: "\pi_A should match \rho_{\pi_E}(s)" --> I think you mean "\rho_{\pi_A} should match \rho_{\pi_E}"?
* L167: "learns only" --> "learns only from"
* Lemma 4.2 RHS, subscript of the expectation: \in should be \sim?

**Questions:**

L147: I don't see why it's a "test time" objective. You would be optimizing your policy with this f-divergence loss, so it comes into play during training time, right?

Theorem 4.1: What does it mean to have a superscript on D_KL? Is this term the same as the LHS of Lemma 4.2?

Do any of the intuitions in this paper transfer to state-only IL, where our dataset doesn't have actions at all? This is becoming an increasingly common setting in the new world of large-scale pre-training (e.g., from human videos).

**Limitations:**

Yes, addressed adequately.

---

> ### Author Rebuttal · Authors · 2023-08-10
>
> We thank the reviewer for their thoughtful review, and we are excited that the reviewer found our insight about the algorithm's role in data quality to be very nice.
>
> **Theoretical Assumptions**
>
> 1. *Why should $\beta$ be small in Lemma 4.2?*
>
> $\beta$ corresponds to the in-distribution loss of the model, and to our knowledge, no imitation learning algorithms would have unbounded in-distribution loss. For example, intervention based methods, adversarial methods, and energy based methods still utilize the training data to model high probability actions. The reviewer states other IL algorithms might sacrifice exactly matching the expert at some states to get better generalization, however once again, to our knowledge, the in-distribution loss is usually still bounded or part of the training objective. We’d be interested in considering any specific algorithms that the reviewer is referring to.
>
> 2. *Gaussian system and policy noise*
>
> We agree with the reviewer that the assumption of Gaussian noise is somewhat simple. We believe that making any further assumptions about the type of system or policy noise would be less general, but this point is well taken and we have included it in our limitations (see one-pager). Modeling and analyzing system noise in real environments is an interesting direction for future work.
>
> **Deterministic Transitions and Policies**
>
> The reviewer raises a concern about obtaining transition diversity in settings with deterministic dynamics. One manner is similar to DART, cited in our paper, where one can artificially inject system noise through the action itself, while learning from the unaltered action. However, in practice, outside of simulated environments we rarely have deterministic dynamics.
>
> The reviewer is correct that 4.1 does not directly apply to deterministic policies, and this is an interesting question. Intuitively, many divergence functions are not ideally suited for comparing two deterministic policies or one stochastic and one deterministic policy, but we might still be able to use naive definitions of “distance” like state MSE in place of the state distribution divergence used in 4.1. However, without more assumptions about the transition function, it is hard to translate state MSE into action MSE to derive a similar theorem to 4.1. Instead, one approximate solution might be to replace the deterministic policy with a unimodal gaussian (with constant variance) for sake of analysis: training a Gaussian with fixed variance with NLL is equivalent to action MSE, and then theorem 4.1 can still apply. Of course, the visited state distribution during deployment (deterministic actions) will be a subset of the visited state distribution used for computing the action divergence (probabilistic actions).
>
> **Maximizing System Entropy**
>
> In the paper we posited that expert demonstrators might encourage paths where the system entropy is higher in order to increase transition diversity. The reviewer is correct that we cannot control the system noise in an environment; however, system noise is conditioned on state and action, and we can still control which part of state/action space we explore during demonstration (i.e., when different parts of state/action space have different system noise). Our claim here is not to increase system noise at a fixed state, but rather that experts can choose different trajectories to demonstrate that are still successful but have more overall system entropy.
>
> **Data size staying fixed in Table 1**
>
> The reviewer states that a simpler explanation for the observed success rates is that BC has less data at each state when data size stays fixed, and thus performance suffers. We clarify that the dataset size is not fixed in terms of number of transitions, but rather number of episodes. Thus longer horizon datasets actually have *more* transitions, and thus should benefit from more state visitation by the same logic. And yet, performance suffers, so we look to action divergence and transition diversity to help understand the results. We will clarify this in the main text, and apologize for the confusion.
>
> **Usage of “Expert” Policy terminology**
>
> The reviewer correctly notes that a true expert would likely not have high action divergence. This is a flaw in our terminology: the expert policy was not meant to imply a perfect demonstrator, and we apologize for the confusion. In the data quality setting, this policy is just the data generating policy, which is defined solely as one that successfully completes the task (this is a more realistic assumption than having access to a true expert). We have updated the paper to reflect this terminology change.
>
> **Distribution shift as a Test time objective**
>
> The objective in Eq.5 is not a training objective, but rather the objective by which we measure the distribution shift at test time. The subtle difference is the order of the f-divergence. We are evaluating the divergence $\rho_\pi$ on the distribution induced by $\rho_\pi$ (test time distribution) not $\rho_{\pi_E}$ (training time distribution).
>
> **Superscript on $D_{KL}$**
>
> Yes this is the same as the LHS of Lemme 4.2. We will expand this out in the text, sorry for the confusion.
>
> **State-only IL**
>
> In state-only IL (e.g. not predicting actions), much of our analysis does not transfer over, since we are primarily dealing with the relationship between actions and states.
>
> Please let us know if you have any further questions or concerns, and in light of our response, we hope that you will consider raising your score.

---

> > ### Comment · Reviewer_9hHD · 2023-08-16
> >
> > Thank you for the detailed author rebuttal and for updating the paper according to reviewer feedback. I am satisfied with the author response, but I maintain my score given my lack of knowledge about the field of imitation learning theory. I would recommend an acceptance if the other, more experienced reviewers do not have a problem with it.

---

### Official Review · Reviewer_24NK · 2023-07-13

**Soundness:** 3 good
**Presentation:** 3 good
**Contribution:** 2 fair
**Rating:** 3
**Confidence:** 4

**Summary:**

The paper addresses a key issue in Imitation Learning (IL) called as distribution shift. The paper formalize data quality and data curation for IL. The paper propose two fundamental properties to evaluate/measure data quality as "action divergence" and "transition divergence". The authors investigate these two properties in context of data quality for IL and presented theoretical and empirical results. They conducted an experiment on human/expert demonstration data and provide analysis on data noising and data measuring.

**Strengths:**

- The paper is well-written
- The idea presented in the paper is very important for IL

**Weaknesses:**

- The work doesn't seems to fit for a high quality conference like NeurIPS
- On measuring the data quality and its effect on success rate, a lot of research is being done on learning from mixed data as its really hard to classify expert vs non-expert data.
- The paper seems more like a white paper or can be submitted to AI Safety conferences but not for NeurIPS
-

**Questions:**

- addressed in Weakness section

**Limitations:**

- Authors are encouraged to explicity mention the limitations of the paper

---

> ### Author Rebuttal · Authors · 2023-08-10
>
> We thank the reviewer for their thoughtful review, and we appreciate that they agree of the importance of the ideas presented in the paper.
>
> **The work doesn't seem to fit for a high quality conference like NeurIPS.**
>
> We disagree that this work is not fit for NeurIPS. As noted by several other reviewers, the question of data quality is an important one for practitioners of imitation learning, which has been largely ignored in recent years. Many people within research and industry  spend countless hours defining clean demonstration strategies, teaching demonstrators to be “proficient,” and filtering data after collection. Yet as a community we do not have a clear understanding of what makes a demonstration clean. In fact, several works submitted in NeurIPS in recent years have used data diversity as a metric for quality [1, 2]. Our work is the first to actually define data quality in imitation learning beyond simplistic notions of “state diversity”, and derive some useful practical insights about state diversity, action divergence, and transition diversity for data curation, which we also show empirically. We value the reviewer’s opinion, and we ask them to elaborate and to provide us with more concrete suggestions for how we can improve our work.
>
> **It’s hard to classify experts vs. non-experts, and thus algorithmic research is more promising**
>
> We completely agree on the difficulty in classifying expert vs. non-expert data; however, in our view this does not make it any less valuable or worth pursuing! We respect the reviewer’s opinion that algorithmic research is more promising, and we hope both avenues can be explored in parallel.
> Recent work in data retrieval [3, 4] have also shown that without changing the algorithm, one can get better policy performance by filtering the data. Therefore, we believe improving dataset quality is an important agenda and with more community focus on data quality, what is considered difficult now will eventually become highly impactful.
>
> **The paper seems more like a white paper or can be submitted to AI Safety conferences**
>
> Our paper includes both theoretical results and empirical analysis that is extremely relevant to the machine learning community, and thus does not fit the definition of a white paper. To add, our primary focus is on improving data quality for final policy performance, and thus we are not focusing on implications of safety in our work. Therefore we do not see how our work would be a fit for an  AI Safety conference.
>
> Please let us know if you have any further questions or concerns, and in light of our response, we hope that you will consider raising your score.
>
> [1] Rajaraman, Nived, et al. "Toward the fundamental limits of imitation learning." Advances in Neural Information Processing Systems 33 (2020): 2914-2924.
>
> [2] Nguyen, Thao, et al. "Quality not quantity: On the interaction between dataset design and robustness of clip." Advances in Neural Information Processing Systems 35 (2022): 21455-21469.
>
> [3] Nasiriany, Soroush, et al. "Learning and retrieval from prior data for skill-based imitation learning." arXiv preprint arXiv:2210.11435 (2022).
>
> [4] Du, Maximilian, et al. "Behavior Retrieval: Few-Shot Imitation Learning by Querying Unlabeled Datasets." arXiv preprint arXiv:2304.08742 (2023).

---

> > ### Comment · Area_Chair_LhJg · 2023-08-19
> > **AC Note for reviewer: Please provide more detail**
> >
> > Dear Reviewer 24NK,
> > I think the weaknesses need to be more about the content of the paper than whether the venue is appropriate or not. Can you please directly provide comments about the content of the paper, otherwise it is challenging to factor the review into the decision. Thank you for your service!

---

### Official Review · Reviewer_zh6Y · 2023-07-16

**Soundness:** 3 good
**Presentation:** 3 good
**Contribution:** 2 fair
**Rating:** 7
**Confidence:** 4

**Summary:**

Data quality is an important aspect of offline robotics because there is not a large quantity of data. In particular, in imitation learning, it has been shown that naive training policies on low-quality data can lead to distribution shift issues. Most researchers focused on dealing with this issue using algorithm-centric approaches. Instead, this paper advocates for a data-centric approach for addressing this issue. The data-centric approach modifies the data collection process, where the data collection can be broken down into two categories: action divergence and transition diversity. The authors formalize these two categories of the data collection process and provide empirical evidence of their significance.

**Strengths:**

Formalizing and addressing the data quality issues in imitation learning is challenging. This paper proposed interesting ideas to this challenging problem and has the following strengths:
- Clarity: The overall paper is well written, and the problem the authors would like to address is clear. The two categories of data collection are also very clear and make it easy to understand the underlying problem dataset attributes.
- Empirical Justification: The author's performance experience using synthetic noise, single and multi-human datasets. All of the experiments performed across these datasets provide insight into various aspects of the data collection discussed in the author's paper. These insights helped enhance the author's discussion of data collection attributes proposed by the authors.

**Weaknesses:**

Although the authors proposed an interesting perspective to address a difficult problem in imitation learning, there exist several weaknesses of the paper:
- Lack of novelty: Theorem 4.1 was already proposed and proven in [1] (see theorem 3 appendix). That means the action divergence interpretation was already discussed in the literature. Furthermore, transition diversity was already proposed and discussed to address distribution shift issues in [2].
- Lack of explanation: The authors emphasize that past work only focuses on state coverage while ignoring that action's role. But if the policy providing the demonstrations is an expert,  then the states provided from the expert should inherently have low action divergence because this is an expert;  and an expert provides the optimal action for a given state.
- Lack of citation: The section on distributions shift in IL needs proper citation.


[1] Imitation Learning as f-Divergence Minimization by Ke et al.
[2] Sequence Model Imitation Learning with Unobserved Contexts

**Questions:**

Below are several questions that I have regarding the proposed idea:
- What is the difference between noise in the expert policy and sub-optimal expert? If there is no difference, then is action divergence focuses on suboptimal experts, not experts?
- In the system noise section, the authors posit that "the expert demosntrators should encourage paths with high system entropy";  how is this concept different than DAgger[1]?
- Why is data quality affected by the algorithm? The algorithm simply uses the data while the expert provides the data. In other words, the algorithms consumes the data while the expert produces the data.
- If the expert is optimal and the transition is not deterministic -- that means that when the expert takes action in state s it will, with some probability end up one of many states, for example. Expert data with a noisy transition function means you get more state coverage. This implies that the expert and policy will have more overlap. What is the difference of this idea and the idea discussed in [2] -- which cover the case when the expert and the learner policy have state distribution overlap.

[1] A Reduction of Imitation Learning and Structured Prediction to No-Regret Online Learning Ross et al.
[2] Feedback in Imitation Learning: The Three Regimes of Covariate Shift Spencer et al.

**Limitations:**

Yes

---

> ### Author Rebuttal · Authors · 2023-08-10
>
> We thank the reviewer for their thoughtful review, and we are glad the reviewer felt our empirical evidence supported our claims.
>
> **Lack of novelty**
>
> The theorem the reviewer cites as similar to 4.1 in [1] looks similar, however it makes the strong assumption of the f-divergence satisfying the triangle inequality, which is not true for the KL divergence we use in 4.1, and also yields a different bound. Furthermore, the implications of the divergence relationship on data quality (i.e. the data generating policy distribution) is not examined within this prior work. They are focusing more on the algorithmic perspective, as is common in prior work. We now explicitly refer to the similarities and differences to this theorem in our discussion. For transition diversity, the concept of transition probabilities is of course not new, however to our knowledge the relationship between state diversity, transition diversity, and action diversity has not been analyzed before to the same extent as our work. We could not find any specific discussion of transition diversity and its role in data quality in the provided citation [2], and so we ask the reviewer to clarify to what specifically they were referring. To our knowledge, [2] instead makes claims about the benefits of state diversity (i.e., low density ratios being beneficial) as is common in prior work.
>
> Furthermore, we believe our conclusion about state diversity is a key contribution: state diversity has been the main metric of “quality” in much of the imitation learning and active learning literature (e.g., when defining a novelty score, or in [2]), and so we strongly believe this point is novel and impactful.
>
> See our main comment response (Novelty).
>
> **Usage of “Expert” Policy terminology**
>
> The reviewer correctly notes that a true expert would likely not have high action divergence. This is a flaw in our terminology: the expert policy was not meant to imply a perfect demonstrator, and we apologize for the confusion. In the data quality setting, this policy is just the data generating policy, which is defined solely as one that successfully completes the task (this is a more realistic assumption than having access to a true expert). We have updated the paper to reflect this terminology change. We hope this also answers the reviewer’s other question about the difference between expert and sub-optimal expert policies.
>
> **Lack of Citation**
>
> We have updated the related work to include [1] and [2], and we are open to any other citations the reviewer believes we should include.
>
> **Difference between System Entropy and DAgger**
>
> We posit that expert demonstrators might encourage paths where the system entropy is higher in order to increase transition diversity (purely offline). DAgger involves querying an expert at robot visited states (online). DAgger makes no claim of which action “strategy” (high or low transition diversity) is better, but rather just relabels any visited states by the robot. In contrast, we are positing that in the offline setting, experts can choose action strategies that give them more coverage over states without increasing action divergence.
>
> **How is data quality (Eq.6) useful if we need to first run the algorithm to evaluate it?**
>
> See our main comment response (same title).
>
> **State Distribution Overlap compared to the Goldilock’s Regime in [2]?**
>
> The high state coverage “Goldilock’s regime” considered in [2] allows them to bound the density ratio between expert and visited state distributions. This is part of a broader class of works that consider the benefit of state diversity, with the loose intuition that state diversity helps us stay in distribution by either expanding the possible successful trajectory space or by introducing recovery behaviors to bring us back in distribution. The difference with our discussion of transition diversity is that we are decoupling action divergence and state diversity. While prior work including [2] broadly consider state diversity as beneficial, the idea that higher state diversity necessarily means we will learn a “recovery” policy is invalid if we have high action divergence. Instead, we look at just how transition diversity can change the state distribution, and thus increase state distribution overlap between expert and policy without incurring high action divergence.
>
> Please let us know if you have any further questions or concerns, and in light of our response, we hope that you will consider raising your score.

---

### Official Review · Reviewer_oFdr · 2023-07-21

**Soundness:** 2 fair
**Presentation:** 3 good
**Contribution:** 3 good
**Rating:** 5
**Confidence:** 2

**Summary:**

This paper systematically studied the data quality assessment problem in imitation learning, proposing new metrics based on test-time distribution shift.


**Strengths:**

The paper is generally well-organized and well-written. The analysis seems to be in fair detail.

**Weaknesses:**

One of the drawn conclusions seem straightforward to me ("increasing state diversity can often come at the expense of action divergence"). Increasing state diversity will bring more challenges for the learning and thus may cause increased action divergence.

The paper lacks practical insights, e.g., how the proposed data quality evaluation metrics can help guide the data collection and thus help the imitation learning is unclear. Currently I dot not see the benefits of introducing such data quality evaluation metrics.

**Questions:**

From Eq.6, the proposed data quality metric is calculated based on the expert policy \pi_E and the learned policy \pi_A by applying an imitation learning algorithm A. If one needs to perform imitation learning first and only after that can get to know how the data quality is, then what is the point of data quality assessment?

Can the authors provide some explanations on how the proposed data quality metrics can be used to benefit the imitation learning?

**Limitations:**

Not discussed in the paper.

---

> ### Author Rebuttal · Authors · 2023-08-10
>
> We thank the reviewer for their thoughtful review, and we appreciate that the reviewer felt our analysis was in fair detail.
>
> **The conclusion on state diversity seems straightforward.**
>
> We agree that this conclusion is intuitive; however, we disagree that it is trivial: state diversity has been the main metric of “quality” in much of the imitation learning and active learning literature (e.g., when defining a novelty score), and so we strongly believe this point is novel and impactful.
>
> See our main comment response (Straightforward Results).
>
> **How can the proposed metrics guide data collection?**
>
> We agree that data quality metrics derived from our theory should eventually be used to guide data collection, and thus we included several practical insights in the “Implications on Data Curation” section (4.3). For example, increasing action consistency during data collection is one potential way to handle action divergence. Furthermore, as mentioned previously, the conclusion on state diversity is extremely practical for data curation. We show the effects of state diversity and action consistency in our experiments. If the reviewer is unclear about how data quality metrics can be useful more generally, we have provided some potential use cases for quality metrics in the future:
>
> 1. To filter datasets and only train on high quality examples without expensive human cleaning effort (which currently requires extensive manual labor and is highly subjective).
> 2. To improve the quality of a given demonstrator online, e.g. through real time feedback like a score function that they can optimize.
> 3. To weight examples during learning.
>
> We also point the reviewer to the Data Measuring experiments in Section 5.2, where we evaluate some initial relevant metrics on various human collected datasets. We see our work as a first step in defining a comprehensive set of data metrics for practitioners to use before training their model.
>
> We hope this provides a broader picture of the utility of data quality metrics in imitation learning.
>
> **How is data quality (Eq.6) useful if we need to first run the algorithm to evaluate it?**
>
> See our main comment response (same title).
>
> Please let us know if you have any further questions or concerns, and in light of our response, we hope that you will consider raising your score.

---

> ### Comment · Reviewer_oFdr · 2023-08-16
>
> I appreciate all the efforts the authors made. I am not familiar with imitation learning, but I am convinced by the authors on their contribution to this field. Thus I changed my score to borderline accept unless other expert reviewers point out any flaws of the paper.

---

### Author Rebuttal · Authors · 2023-08-10

We thank all the reviewers for their detailed and constructive feedback. We are excited that the majority of reviewers agree about the importance of the data quality problem in imitation learning and find our analysis sound.

We outline the points of important feedback that came up several times in the reviews below:

**Novelty**

We apologize for any confusing presentation of the work leading some reviewers to fail to see the contributions of our work. Our contributions cannot be properly assessed without understanding the broad literature of current developments in machine learning and the history of IL research. Recently there has been a paradigm shift in NLP and CV from developing large models to curating high-quality datasets, and we believe robotics can learn from these lessons as we are starting to see increasingly large models. A body of IL research relied on near-optimal or machine-generated datasets before the community realized machine-generated data is not representative of human data [1]. The community largely has chosen to collect large amounts of human data and hope algorithmic fixes can overcome the challenges of uncurated data, for example with large transformers like RT-1 [2]. However, as the results of these prior works show, algorithmic changes without close attention to data quality can only take us so far. Instead our work shows that a more efficient and principled approach is to analyze what properties of machine-generated data allow simple methods to work and how to curate human-generated data to be friendly to existing IL models.

We would like to re-emphasize the core contribution of our work: we are the first to formalize data quality in imitation learning in an effort to guide data curation in practice. Our work is a first step towards this paradigm shift in the IL community and we hope the reviewers can see the importance of this message. In addition, our work presents two key insights that challenge common beliefs of the IL community:
- the downstream learning algorithm is an important factor when assessing data quality
- state diversity alone is not sufficient for measuring data quality

Empirically, we study how action divergence and transition diversity in a dataset influence downstream performance in controlled settings. While our analysis informs what properties we want a dataset to have (i.e., higher transition diversity up to a point, lower action divergence), we also show the difficulty in designing universal data quality metrics, opening an important problem for the community.

**Straightforward Results**

Two reviewers found some results to be straightforward. While we agree several of the conclusions are intuitive, we disagree that they are trivial of straightforward: state diversity has been the main metric of “quality” in much of the imitation learning and active learning literature, and so we strongly believe this point – going beyond simply optimizing for state diversity as done in prior work – is novel and impactful [3,4,5].

Furthermore, simplicity is not the same as significance. The impact of theoretical results depends on how well they explain practical results, not how simple the proofs are to understand. Our theoretical claims are actually supported by the practical results on several environments, and as exemplified above this could have high significance in the field of imitation learning.


**How is data quality (Eq.6) useful if we need to first run the algorithm to evaluate it?**

To be clear, the definition in Eq.6 is a theoretical definition of data quality, and acts more as a north star rather than a tractable metric. Our work is the first to define and study quality at a theoretical level. We spend the rest of the theoretical section using this formula to derive more tractable metrics like action consistency, horizon length, and transition diversity, which do not require an algorithm to evaluate. This question is precisely what makes analyzing data quality so difficult, and our work is the first to recognize this relationship between data quality and the algorithm, formalize it, and analyze it in practice. Our experiments in 5.1 and 5.2 show how this theory can be useful in practice: by understanding properties of a given algorithm *class* (e.g. the benefits of action consistency for behavior cloning methods), we can bypass the need for model training to evaluate data quality.

**Full list of changes and updates:**
- Discussion of fairness and safety implications for data quality metrics (see one-pager)
- Adding a limitations section for theoretical assumptions, metric limitations, etc (see one-pager).
- Adding system noise + policy noise experiments for Square (see one-pager)
- Adding transformer based experiments to Appendix (see one-pager)
- Adding results on system and policy noise without multimodality in PMObstacle
- Motivating how we would use metrics to guide data collection in the text
- Changing “expert” to “demonstrator” policy
- Including missing citations suggested by Reviewer 2 (zh6Y)
- Clarifying dataset details for Section 5.2, and longer analysis of Metrics in Section 5.2
- Rewriting the KL divergence using the same notation as the left hand side of Lemma 4.2
- Various typos presented by the reviewers
- Better explanation of the experiment figure and better labeling

[1] Mandlekar, Ajay, et al. "What matters in learning from offline human demonstrations for robot manipulation.".

[2] Brohan, Anthony, et al. "Rt-1: Robotics transformer for real-world control at scale.".

[3] Judah, Kshitij, et al. "Active lmitation learning: formal and practical reductions to IID learning.".

[4] Hawke, Jeffrey, et al. "Urban driving with conditional imitation learning.".

[5] Antotsiou, Dafni, Carlo Ciliberto, and Tae-Kyun Kim. "Adversarial imitation learning with trajectorial augmentation and correction.".

---

### Decision · Program_Chairs · 2023-09-21

**Decision:**

Accept (poster)

**Comment:**

The authors propose a method to understand the qualities of a *good* dataset for imitation learning, for the purposes of data curation. They find interesting insights regarding the transition diversity and action divergence being good metrics to measure dataset quality. They pair a more formal description of these phenomena with a variety of empirical analyses on imitation learning data in simulated robotics. The resulting paper provides a set of actionable insights for practitioners of imitation learning.

During the review period, there were 2 very uninformative reviews, which we discarded during the discussion period. All the other reviewers find the insights described quite useful and the formalism introduced will be useful for future imitation learning practitioners. The reviewers brought up valuable points of clarification to be added to the paper, please do incorporate them for a camera-ready version.